

# Argyres-Douglas theories in class $\mathcal{S}$ without irregularity

**Christopher Beem$^{\star}$ and Wolfger Peelaers$^{\dagger}$**

Mathematical Institute, University of Oxford,
Woodstock Road, Oxford, OX2 6GG, United Kingdom

$\star$ christopher.beem@maths.ox.ac.uk , $\dagger$ wolfger.peelaers@maths.ox.ac.uk

## Abstract

We make a preliminary investigation into twisted $A_{2n}$ theories of class $\mathcal{S}$. Contrary to a common piece of folklore, we establish that theories of this type realise a variety of models of Argyres-Douglas type while utilising only regular punctures. We present an in-depth analysis of all twisted $A_2$ trinion theories, analyse their interrelations via partial Higgsing, and discuss some of their generalised S-dualities.



# 1 Introduction and summary

The ecosystem of four-dimensional $\mathcal{N} = 2$ superconformal field theories (SCFTs) is remarkably diverse. A particularly abundant species of such theories are those of *class* $\mathcal{S}$. Discovered over a decade ago in [1, 2], their characterisation as partially twisted compactifications of a six-dimensional $\mathcal{N} = (2, 0)$ theory on a punctured Riemann surface allows for a straightforward taxonomy: a theory is fully specified by a choice of a simply-laced Lie algebra $\mathfrak{j} = \mathfrak{a}_n, \mathfrak{d}_n$, or $\mathfrak{e}_{6,7,8}$ labelling the parent $\mathcal{N} = (2, 0)$ theory, a Riemann surface of genus $g$ with $s$ marked points, and a choice of $s$ half-BPS, codimension-two defects to be inserted at said marked points. For a particularly nice set of defects, referred to as *regular* or *tame*, the latter choice amounts to specifying an embedding $\mathfrak{su}(2) \hookrightarrow \mathfrak{j}$ up to conjugacy. Furthermore, one can include codimension-two *twisted* defects (also called *monodromy defects*), which introduce a monodromy by an element of the outer automorphism group of $\mathfrak{j}$ when encircled [3–5]. These are labelled by an embedding $\mathfrak{su}(2) \hookrightarrow \mathfrak{g}$ where $\mathfrak{g} = \mathfrak{j}_0^\vee$ is the Langlands dual of the invariant subalgebra $\mathfrak{j}_0 \subset \mathfrak{j}$ with respect to the action of the outer automorphism.[1] Amongst the theories of class $\mathcal{S}$, the elementary taxa are those associated with three-punctured spheres—all other theories can be obtained by exactly marginal gaugings thereof. These theories are often referred to as *trinion theories*. The richness of class $\mathcal{S}$ stems largely from the wide variety of allowed choices of triples of embeddings labelling their punctures. In a series of papers [5–14], trinion theories of all types $\mathfrak{j}$ with or without twisted punctures were scrutinised, with the exception of theories of type $\mathfrak{a}_{2n}$ in the presence of $\mathbb{Z}_2$-twisted punctures. This paper should motivate a more systematic study of this last class of theories.

Twisted $A_{2n}$ theories have not yet been subjected to a methodical investigation due to difficulties that are expected to arise in such an endeavour due to the subtleties identified in [15]. Consequently, very few results have been established up to now. In brief, the results of investigations of which the authors are aware are as follows:[2]

(i) By means of an $S$-duality, the authors of [8] argued that the four-dimensional $\mathcal{N} = 2$ SCFT with a one-complex-dimensional Coulomb branch and flavour symmetry $C_2 \times U(1)$, whose existence had previously been proposed in [17, 18], can be realised as a twisted $A_2$ theory.

(ii) This theory and its higher-rank analogues were further analysed in [19] at the level of their superconformal indices and Higgs branch chiral rings.

(iii) It was explained in [20] that the $USp(2n)$ flavour symmetry of a full, twisted $A_{2n}$ puncture carries a Witten anomaly.

In this paper we find that these theories may actually be investigated and understood quite effectively. By bringing to bear an array of robust diagnostics—to wit, the $a$ and $c$ conformal anomaly coefficients, flavour symmetries and their associated central charge, the superconformal index, the Higgs branch chiral ring, and the associated vertex operator algebra—we analyse in great detail the complete family of twisted $A_2$ trinion theories. Table 1 summarises some of their properties. We find that all but one of these trinion theories can be identified with well-studied SCFTs. Strikingly, several members of the twisted $A_2$ family are Argyres-Douglas models, *i.e.*, their Coulomb branch chiral rings include generators with non-integer scaling dimensions,[3] thus refuting an oft-repeated piece of folklore that states that class $\mathcal{S}$ the-

---

[1]Also, when twisted defects are present, a selection rule/consistency condition demands that they are present in appropriate multiples. In particular, for $\mathbb{Z}_2$ twists they should come in even numbers.

[2]An analysis of irregular, twisted defects, including $A_{2n}$ monodromy defects, was carried out in [16]. See also footnote 4.

[3]The name of these theories refers to the authors of the paper [21], in which the first such models were discovered. See also [22].

Table 1: Twisted $A_2$ trinions. Gray punctures represent monodromy defects and are connected by a $\mathbb{Z}_2$-twist line. The number of Coulomb branch chiral ring generators of scaling dimension $i$ is denoted by $d_i$. The $a$ and $c$ central charges and the flavour symmetry groups $G_F$ of (the interacting parts of) the theories are indicated as well. The subscript on the flavour symmetry groups denotes the corresponding flavour central charge. In the identification of the theory described by the trinion, we denote the rank-$n$ $\mathfrak{su}(3)$-instanton SCFT as $\mathcal{T}_{\mathfrak{su}(3)}^{(n)}$. In particular, $\mathcal{T}_{\mathfrak{su}(3)}^{(1)}$ is the $(A_1, D_4)$ Argyres-Douglas theory. Finally, HM stands for hypermultiplet.

| | | | |
|---|---|---|---|
| | | | |
| $(d_{\frac{3}{2}}, d_3)$; $(a,c)$ | $(0,2)$; $(\frac{11}{4}, 3)$ | $(1,1)$; $(\frac{47}{24}, \frac{13}{6})$ | $(2,0)$; $(\frac{7}{6}, \frac{4}{3})$ |
| $G_F$ | $SU(3)_6 \times (SU(2)_4)^2$ | $SU(3)_6 \times SU(2)_4$ | $SU(3)_3 \times SU(3)_3$ |
| Description | $\widetilde{T}_3$ | $\mathcal{T}_{\mathfrak{su}(3)}^{(2)}$ | $\left(\mathcal{T}_{\mathfrak{su}(3)}^{(1)}\right)^{\otimes 2}$ |
| | | | |
| $(d_{\frac{3}{2}}, d_3)$; $(a,c)$ | $(0,1)$; $(\frac{17}{12}, \frac{19}{12})$ | $(1,0)$; $(\frac{7}{12}, \frac{2}{3})$ | |
| $G_F$ | $USp(4)_4 \times U(1)$ | $SU(3)_3$ | |
| Description | rank-one $C_2 U_1$ theory | $\mathcal{T}_{\mathfrak{su}(3)}^{(1)} \otimes$ HM | |

ories constructed with regular punctures can only accommodate integer scaling dimensions. In other words, one need not introduce *irregular* (also called *wild*) punctures to construct Argyres-Douglas theories in class $\mathcal{S}$.[4] On the other hand, the evidence before us suggests that while twisted $A_{2n}$ theories allow for half-integer Coulomb branch scaling dimensions, more general fractions cannot be realised.

The class $\mathcal{S}$ realisations of Table 1 expose a rich network of connections via partial Higgsing operations. Indeed, the operation of partially closing a puncture has long been recognised as a partial Higgsing triggered by a nilpotent vacuum expectation value for the moment map operator associated with the flavour symmetry carried by that puncture [5, 25–27]. Nevertheless, these partial Higgsings appear surprising when phrased in terms of the SCFTs identified with the trinion theories; in fact, they anticipate and confirm instances of novel interrelations between $\mathcal{N} = 2$ SCFTs studied in [28–30]. Leveraging the recent insights of [31], the Higgsing in these examples can be "undone" at the level of the Higgs branch geometry and associated

---

[4]See, for example, [16, 23, 24] for a systematic analysis of class $\mathcal{S}$ constructions of Argyres-Douglas theories involving irregular punctures.

vertex operator algebra, providing an efficient and effective tool to construct these quantities for the un-Higgsed theories.

A central property of theories of class $\mathcal{S}$ is that their various (generalised) $S$-duality frames are manifested geometrically as different degeneration limits of the corresponding punctured Riemann surfaces (their *UV curves*). For twisted $A_2$ theories this allows us to establish a number of $S$-duality relations involving rank-one and rank-two $\mathfrak{su}(3)$-instanton SCFTs. In particular, the $S$-duality studied in [32] is easily confirmed this way.

The remainder of this paper is organised as follows. In Section 2, we recall essential aspects of the class $\mathcal{S}$ construction and review the determination of the various key quantities available to us for the analysis of such theories. In section 3 we leverage these tools to identify all twisted $A_2$ theories. We study the interrelations via partial Higgsings among these theories in Section 4, while in Section 5 we detail some paradigmatic $S$-dualities. Finally, in Section 6, we comment on future directions for further study and present some motivational results in these directions. We identify the entire infinite series of $D_2[SU(2n+1)]$ Argyres-Douglas fixed points as twisted $A_{2n}$ theories, and make a proposal for how the half-integer scaling dimensions in these models arise in this setting.

## 2  $\mathcal{N} = 2$ SCFTs of class $\mathcal{S}$

In this section, we briefly recall several key aspects of the class $\mathcal{S}$ construction of four dimensional $\mathcal{N} = 2$ superconformal field theories. We focus on the irreducible class $\mathcal{S}$ objects, the trinion theories, and review universal formulae for their Weyl anomaly coefficients $a$ and $c$, and their flavour central charges. We recall the general, TQFT expression for the Macdonald limit of their superconformal indices and briefly discuss the realisation of the SCFT/VOA correspondence [33] in this setting. Readers familiar with the class $\mathcal{S}$ literature, salient features of which were reviewed in [27,34], may safely skip this section.

Theories of class $\mathcal{S}$ were introduced in [1,2] and are most usefully thought of as the low-energy limits of (partially) twisted compactifications of a six-dimensional $\mathcal{N} = (2,0)$ superconformal field theory on a Riemann surface. This setup is usually enriched with half-BPS, codimension-two defects of the six-dimensional theory located at marked points on the Riemann surface and spanning the four non-compact spacetime dimensions. The defining data of the resulting four-dimensional superconformal field theory is thus as follows:

1. A simply-laced Lie algebra $\mathfrak{j} \in \{\mathfrak{a}_n, \mathfrak{d}_n, \mathfrak{e}_{6,7,8}\}$, which labels the above-lying six-dimensional $\mathcal{N} = (2,0)$ superconformal field theory.[5]

2. A Riemann surface $\mathcal{C}_{g,s}$ of genus $g$ with $s$ marked points. Of the geometric data for the surface, only the complex structure moduli are retained in the low-energy limit—they correspond to exactly marginal couplings of the four-dimensional SCFT.[6]

3. For each of the $s$ marked points, a half-BPS, codimension-two defect. One often restricts to *regular* or *tame* defects, which are labelled by an embedding $\Lambda : \mathfrak{su}(2) \hookrightarrow \mathfrak{j}$. One can further consider defects that introduce a monodromy by an element $\sigma$ of the outer-automorphism group of $\mathfrak{j}$ when encircled [3–5]. These *twisted defects* are labelled by an embedding $\mathfrak{su}(2) \hookrightarrow \mathfrak{g}$, where $\mathfrak{g} = \mathfrak{j}_0^{\vee}$ is the Langlands dual of the $\sigma$-invariant subalgebra $\mathfrak{j}_0 \subset \mathfrak{j}$, see Table 2 for a list of cases. From twisted punctures emanate topological

---

[5]This ADE classification is, for example, manifest in the realisation of the $(2,0)$ theory as the low-energy limit of type IIB string theory on the corresponding ALE space [35].

[6]The theory may possess additional exactly marginal couplings at values frozen in the interior of their moduli space [8].

Table 2: Simple Lie algebras $\mathfrak{j}$ whose Dynkin diagrams possess nontrivial discrete symmetry groups generated by an element $\sigma$, the subgroups of the outer automorphism group $\langle\sigma\rangle$ that is generated by $\sigma$, the $\sigma$-invariant subalgebras $\mathfrak{j}_0 \subset \mathfrak{j}$, and the Langlands dual $\mathfrak{g} = \mathfrak{j}_0^\vee$.

| $\mathfrak{j}$ | $\mathfrak{a}_{2n-1}$ | $\mathfrak{a}_{2n}$ | $\mathfrak{d}_n$ | $\mathfrak{d}_4$ | $\mathfrak{e}_6$ |
|---|---|---|---|---|---|
| $\langle\sigma\rangle$ | $\mathbb{Z}_2$ | $\mathbb{Z}_2$ | $\mathbb{Z}_2$ | $\mathbb{Z}_3$ | $\mathbb{Z}_2$ |
| $\mathfrak{j}_0$ | $\mathfrak{c}_n$ | $\mathfrak{b}_n$ | $\mathfrak{b}_{n-1}$ | $\mathfrak{g}_2$ | $\mathfrak{f}_4$ |
| $\mathfrak{g}$ | $\mathfrak{b}_n$ | $\mathfrak{c}_n$ | $\mathfrak{c}_{n-1}$ | $\mathfrak{g}_2$ | $\mathfrak{f}_4$ |

twist lines, which are codimension-one topological defect operators of the parent six-dimensional theory. These keep track of the monodromies on the UV curve and can be used to ensure that a given set of twisted defects is globally allowable, as well as affecting the interpretation of degeneration limits when a twist line passes through the degeneration.

Any Riemann surface $\mathcal{C}_{g,s}$ admits a variety of pants decompositions that deconstruct the surface into $2g+s-2$ three-punctured spheres glued together by connecting $3g+s-3$ pairs of punctures. Per the second item above, each such decomposition corresponds to a degeneration limit in the complex structure moduli space of the UV curve, which in turn corresponds to a regime in coupling space of the corresponding four-dimensional superconformal field theory described by weakly coupled, exactly marginal gaugings of the SCFTs associated with the three-punctured spheres. Generalised $S$-duality relates the different weakly coupled descriptions corresponding to the different pants decompositions of $\mathcal{C}_{g,s}$.

The study of theories of class $\mathcal{S}$ thus starts with the project of coming to grips with the isolated, usually non-Lagrangian, *trinion theories* associated with three-punctured spheres. This herculean task has been largely completed in a sequence of papers [5–14]. These papers describe the trinions for all twisted and untwisted classes of theories with the exception of *twisted $A_{2n}$ models*. The latter are the subject of this paper.

## 2.1 Trinion theories

Trinion theories are the basic building blocks of class $\mathcal{S}$. Apart from the choice of simply-laced Lie algebra $\mathfrak{j}$, they are specified by three embeddings $\Lambda_i : \mathfrak{su}(2) \hookrightarrow \mathfrak{j}$, $i = 1, 2, 3$, or in the twisted case, $\Lambda_i : \mathfrak{su}(2) \hookrightarrow \mathfrak{g}$. To uniformise the discussion, we will henceforth set $\mathfrak{g} = \mathfrak{j}$ for untwisted punctures. We denote trinion theories as $T_{\mathfrak{j}}^{\Lambda_1\Lambda_2\Lambda_3}$, where the choice of embeddings specifies if the theory is twisted or untwisted.[7]

A puncture labelled by the embedding $\Lambda$ contributes to the flavour symmetry of the trinion theory a factor $\mathfrak{f}_\Lambda \subset \mathfrak{g}$ given by the commutant of the image of the embedding. The flavour symmetry of $T_{\mathfrak{j}}^{\Lambda_1\Lambda_2\Lambda_3}$ thus contains *at least* the algebra $\oplus_i \mathfrak{f}_{\Lambda_i}$, which, in exceptional cases, may be further enhanced. We will present a diagnostic for such enhancements below (see below (2.13)).

For the computational recipes below, it will be useful to introduce notation for the decom-

---

[7]Not all choices of triples of embeddings correspond to physical, four-dimensional SCFTs. A diagnostic to detect disallowed triples is to check the dimensions of the graded components of the Coulomb branch (viewed as a graded vector space) of the putative theory using the algorithms of [5–14]. If any of them is negative, the theory is unphysical. Alternatively, if the expression for the superconformal index presented below in (2.9) diverges, the theory is designated as bad [36].

position of the adjoint representation of $\mathfrak{g}$ under the subalgebra $\Lambda(\mathfrak{su}(2)) \oplus \mathfrak{f}_\Lambda$,

$$\mathfrak{g} = \bigoplus_{j \in \frac{1}{2}\mathbb{Z}_{\geqslant 0}} V_j \otimes \mathcal{R}_j \,, \tag{2.1}$$

where $V_j$ is the spin-$j$ representation and $\mathcal{R}_j$ is some (possibly reducible, possibly zero- dimensional) representation of $\mathfrak{f}_\Lambda$.

## 2.2 Central charges

The conformal anomaly coefficients $a$ and $c$ of $T_j^{\Lambda_1 \Lambda_2 \Lambda_3}$ are conventionally expressed in terms of the effective number of vector multiplets ($n_v$) and hypermultiplets ($n_h$),

$$a = \frac{2n_v + n_h}{12} \,, \qquad c = \frac{5n_v + n_h}{24} \,. \tag{2.2}$$

These are then given in terms of our class $\mathcal{S}$ data by [5]

$$n_v = \sum_{i=1}^{3} n_v(\Lambda_i) - (\tfrac{4}{3} h_{\mathfrak{j}}^\vee \dim \mathfrak{j} + \operatorname{rank} \mathfrak{j}) \,, \qquad n_h = \sum_{i=1}^{3} n_h(\Lambda_i) - \tfrac{4}{3} h_{\mathfrak{j}}^\vee \dim \mathfrak{j} \,, \tag{2.3}$$

where

$$n_v(\Lambda) = 8(\tfrac{1}{12} h_{\mathfrak{j}}^\vee \dim \mathfrak{j} - \rho_{\mathfrak{g}} \cdot \tfrac{h}{2}) + \tfrac{1}{2}(\operatorname{rank} \mathfrak{j} - \dim \mathfrak{g}_0) \,, \tag{2.4}$$

$$n_h(\Lambda) = 8(\tfrac{1}{12} h_{\mathfrak{j}}^\vee \dim \mathfrak{j} - \rho_{\mathfrak{g}} \cdot \tfrac{h}{2}) + \tfrac{1}{2} \dim \mathfrak{g}_{1/2} \,. \tag{2.5}$$

Here $h_{\mathfrak{j}}^\vee$ is the dual Coxeter number of $\mathfrak{j}$ and $\rho_{\mathfrak{g}}$ is the Weyl vector (*i.e.*, half the sum of the positive roots) of $\mathfrak{g}$. (Recall that if the defect is untwisted, we set $\mathfrak{g} = \mathfrak{j}$.) Furthermore, we have $h := \Lambda(\sigma_3)$, the image of the Cartan element of $\mathfrak{su}(2)$, and the formulae also involve the quantities $\dim \mathfrak{g}_0$ and $\dim \mathfrak{g}_{1/2}$ defined as

$$\dim \mathfrak{g}_0 := \sum_{j \in \mathbb{Z}_{\geqslant 0}} \dim \mathcal{R}_j \,, \qquad \dim \mathfrak{g}_{1/2} := \sum_{j \in \frac{1}{2} + \mathbb{Z}_{\geqslant 0}} \dim \mathcal{R}_j \,. \tag{2.6}$$

These can be thought of as the dimensions of the $\frac{1}{2}h$-eigenspaces of eigenvalues 0 and 1/2 respectively.

A simple factor $\mathfrak{f}' \subset \mathfrak{f}_\Lambda$ has flavour central charge [5]

$$k_{\mathfrak{f}'} = 2 \sum_{j \in \frac{1}{2}\mathbb{Z}_{\geqslant 0}} T(\mathcal{R}_j|_{\mathfrak{f}'}) \,, \tag{2.7}$$

where $T(\mathcal{R})$ denotes the Dynkin index of the representation $\mathcal{R}$.[8] The representations $\mathcal{R}_j$ are the ones appearing in the decomposition (2.1), and we are treating them as (potentially reducible) $\mathfrak{f}'$ representations.[9] In particular, the level of the flavour symmetry $\mathfrak{g}$ of a puncture labelled by the trivial embedding, often called *a full puncture*, is given by twice the dual Coxeter number of $\mathfrak{g}$.

---

[8] The normalisation is such that the index of the adjoint representation equals the dual Coxeter number, *i.e.*, $T(\mathfrak{g}) = h_{\mathfrak{g}}^\vee$, for any Lie algebra $\mathfrak{g}$.

[9] Concretely, if $\mathfrak{f}_\Lambda = \mathfrak{f}' \oplus \hat{\mathfrak{f}}'$ for some (not necessarily simple) complementary factor $\hat{\mathfrak{f}}'$ and $\mathcal{R}$ contains an $n$-fold degenerate representation $r \otimes \hat{r}$, then $\mathcal{R}|_{\mathfrak{f}'}$ contains the $(n \dim \hat{r})$-fold degenerate representation $r$.

Table 3: List of outer-automorphism invariant representations of $\mathfrak{j}$, specified by their highest weight state $\lambda_{\mathfrak{j}}$, determined by representations $\lambda_{\mathfrak{g}} = \sum_{i=1}^{\text{rank}\,\mathfrak{g}} \lambda_i \omega_i$ of $\mathfrak{g}$, the Langlands dual of the outer-automorphism invariant subalgebra. The highest weight states are expressed in a basis of fundamental weights $\omega_i$ (the coefficients are the Dynkin labels of the representation) for which we follow the conventions of LieArt [43, 44].[9]

| $\mathfrak{g}$ | $\mathfrak{j}$ | $\lambda_{\mathfrak{j}}$ |
|---|---|---|
| $\mathfrak{b}_n$ | $\mathfrak{a}_{2n-1}$ | $\sum_{i=1}^{n} \lambda_i \omega_i + \sum_{i=1}^{n-1} \lambda_{n-i} \omega_{n+i}$ |
| $\mathfrak{c}_n$ | $\mathfrak{a}_{2n}$ | $\sum_{i=1}^{n} \lambda_i \omega_i + \sum_{i=1}^{n} \lambda_{n+1-i} \omega_{n+i}$ |
| $\mathfrak{c}_n$ | $\mathfrak{d}_{n+1}$ | $\sum_{i=1}^{n} \lambda_i \omega_i + \lambda_n \omega_{n+1}$ |
| $\mathfrak{g}_2$ | $\mathfrak{d}_4$ | $\lambda_2 \omega_1 + \lambda_1 \omega_2 + \lambda_2 \omega_3 + \lambda_2 \omega_4$ |
| $\mathfrak{f}_4$ | $\mathfrak{e}_6$ | $\lambda_1 \omega_1 + \lambda_2 \omega_2 + \lambda_3 \omega_3 + \lambda_2 \omega_4 + \lambda_1 \omega_5 + \lambda_4 \omega_6$ |

## 2.3  Superconformal index

The *Macdonald limit* of the superconformal index of a four-dimensional $\mathcal{N} = 2$ superconformal field theory is defined as [37]

$$I_M(q, t; \mathbf{a}) = \text{tr}_M \, (-1)^F q^{E-2R+r} t^{R-r} \prod_i a_i^{f_i} \,, \qquad (2.8)$$

where the trace runs over states in the Hilbert space of the radially quantised SCFT satisfying $E - (j_1 + j_2) - 2R = 0$ and $j_1 - j_2 + r = 0$. Here $E$ denotes the conformal dimension, $(j_1, j_2)$ are the Cartans of the $\mathfrak{su}(2)_1 \oplus \mathfrak{su}(2)_2$ rotational group, $(R, r)$ are the Cartan elements of the R-symmetry algebra $\mathfrak{su}(2)_R \oplus \mathfrak{u}(1)_r$, and $f_i$ are a basis of flavour symmetry Cartan generators. The index is independent of exactly marginal couplings and hence, for theories of class $\mathcal{S}$, it is computed by a topological quantum field theory on $\mathcal{C}_{g,s}$ [38]. The relevant TQFT for the index (2.8) has been identified as $(t, q)$-deformed two-dimensional Yang-Mills theory in the zero-area limit. The Macdonald index of a trinion theory $T_{\mathfrak{j}}^{\Lambda_1 \Lambda_2 \Lambda_3}$ is computed by that TQFT's structure constants and reads [37, 39–42]

$$I_M^{\mathfrak{j};\Lambda_1 \Lambda_2 \Lambda_3}(q, t; \mathbf{a}_i) = \sum_\lambda \frac{\prod_{i=1}^{3} K_{\Lambda_i}(\mathbf{a}_i) \, P_\lambda^{\mathfrak{g}_i}(t^{\frac{1}{2}\Lambda_i(\sigma_3)} \otimes \mathbf{a}_i)}{K_\rho \, P_\lambda^{\mathfrak{j}}(t^{\frac{1}{2}\rho(\sigma_3)})} \,. \qquad (2.9)$$

Let us unpack this expression.

- In the untwisted case, the sum runs over all finite-dimensional representations $\lambda$ of $\mathfrak{j}$. If any of the punctures is twisted, then the sum instead runs over representations of the algebra $\mathfrak{g}$ associated with the twisted punctures (which come in pairs for $\mathbb{Z}_2$-twists and in pairs or triples for $\mathbb{Z}_3$-twists). In this case, it will be useful to associate with each representation $\lambda$ of $\mathfrak{g}$ an outer-automorphism invariant representation of $\mathfrak{j}$ determined by the same Dynkin labels, see Table 3. In a slight abuse of notation, we denote that representation again by $\lambda$. In either case, we continue with the convention that for untwisted punctures $\mathfrak{g} = \mathfrak{j}$.

- In the summand one encounters $P_\lambda^{\mathfrak{g}/\mathfrak{j}}$, which are unconventionally normalised Macdonald polynomials associated to the representations $\lambda$ of $\mathfrak{g}/\mathfrak{j}$ [45]. The polynomials $P_\lambda$ are orthonormal under the measure

$$\frac{(q;q)^r}{(t;q)^r}[d\mathbf{x}]_M = \mathrm{PE}\left[\frac{-q+t}{1-q}\chi_{\mathrm{adj}}(\mathbf{x})\right][d\mathbf{x}]\,, \tag{2.10}$$

where $[d\mathbf{x}]_M$ is the Macdonald measure and $[d\mathbf{x}]$ denotes the Haar measure.[10] Furthermore, $r$ denotes the rank of the relevant Lie algebra and $\chi_{\mathrm{adj}}$ is its adjoint character. We also used the infinite $q$-Pochhammer symbol $(a;q) = \prod_{j=0}^\infty(1-aq^j)$ and the plethystic exponential $\mathrm{PE}[f(x_i)] = \exp(\sum_{n=1}^\infty \frac{1}{n}f(x_i^n))$.

The argument of the polynomials is a $\mathrm{Lie}(\mathfrak{g}/\mathfrak{j})$ group element (conjugated into its maximal torus). In the numerator, reflecting the decomposition $\Lambda(\mathfrak{su}(2)) \oplus \mathfrak{f}_\Lambda \subset \mathfrak{g}$, it is obtained as the direct product of an $SU(2)$ group element—the exponentiated Cartan generator—and a $\mathrm{Lie}(\mathfrak{f}_\Lambda)$ element. In the denominator, its argument is determined similarly in terms of the principal embedding $\rho$.

- Finally, again in terms of (2.1), the "K-factors" are given by

$$K_\Lambda(\mathbf{a}) = \mathrm{PE}\left[\sum_j \frac{t^{1+j}}{1-q}\chi_{\mathcal{R}_j}(\mathbf{a})\right]\,, \tag{2.11}$$

where $\chi_{\mathcal{R}}$ denotes the character of the representation $\mathcal{R}$.

The Macdonald index provides a straightforward method to determine the number of free hypermultiplets a theory contains and to find out if the flavour symmetry of a theory is enhanced. Expanding to order $O(t,q)$, one finds

$$I_M = 1 + \chi_{2n}t^{\frac{1}{2}} + \chi_{\mathrm{adj}}t + \dots\,. \tag{2.13}$$

Here $\chi_{2n}$ is the (possibly not fully refined) character of the fundamental representation of $\mathfrak{usp}(2n)$. If $n > 0$, the theory contains $n$ free hypermultiplets. It is straightforward to probe the index of only the interacting part of the theory by dividing out the contribution of these free multiplets,

$$\tilde{I}_M = \mathrm{PE}\left[-\frac{t^{\frac{1}{2}}}{1-q}\chi_{2n}\right]I_M\,. \tag{2.14}$$

Furthermore, $\chi_{\mathrm{adj}}$ is the (possibly not fully refined) character of the adjoint representation of the flavour symmetry group of the theory. In the presence of free hypermultiplets it can be written as $\chi_{\mathrm{adj}} = \tilde{\chi}_{\mathrm{adj}} + \chi_{\mathrm{adj}_{\mathfrak{usp}(2n)}}$ where $\tilde{\chi}_{\mathrm{adj}}$ captures the flavour symmetry of the interacting part of the theory, i.e., it is the coefficient of $t$ of $\tilde{I}_M$. For trinion theories, one finds $\chi_{\mathrm{adj}} = \sum_i \chi_{\mathrm{adj}_{\mathfrak{f}_{\Lambda_i}}} + \dots$, and a nonempty ellipsis signals an enhancement of the manifest flavour symmetry $\oplus_i \mathfrak{f}_{\Lambda_i}$.

An important further limit of the index (2.8) is $t \to q$. This limit is known as the *Schur limit*, because the Macdonald polynomials in (2.9) simplify to characters, also known as Schur

---

[9]Unfortunately, these differ from the more established conventions of Bourbaki [46].

[10]Concretely,

$$[d\mathbf{x}]_M = \prod_j \frac{dx_j}{2\pi i x_j}\prod_{\alpha\neq 0}\frac{(e^\alpha;q)}{(te^\alpha;q)}\,, \qquad [d\mathbf{x}] = \prod_j \frac{dx_j}{2\pi i x_j}\prod_{\alpha\neq 0}(1-e^\alpha)\,, \tag{2.12}$$

where the products run over the non-zero roots of the Lie algebra. The fugacities $x_j$ can be identified as the exponentials $x_j = e^{b_j}$ of the elements $b_j$ of a basis of weights. Standard choices are the basis of fundamental weights $\omega_i$ or orthogonal weights $\epsilon_i$.

polynomials. Its particular usefulness stems from the fact that in this limit, the index takes the form[11]

$$I_M(q,q;\mathbf{a}) = I_S(q;\mathbf{a}) = \text{tr}\,(-1)^F q^{E-R} \prod_i a_i^{f_i}\,, \qquad (2.15)$$

which in particular means that the Schur index equals the vacuum character of the associated vertex operator algebra of the four-dimensional superconformal field theory (up to a Casimir prefactor) [33]. This equality is just one facet of the SCFT/VOA correspondence, to which we briefly turn below. We also note that the limit $q \to 0$ of the Macdonald index returns the *Hall-Littlewood limit* of the index. For trinion theories, this quantity equals the Hilbert series of the Higgs branch chiral ring of the theory *i.e.*, the ring of operators characterised by $E = 2R$ [37].

## 2.4 SCFT/VOA correspondence

It was shown in [33], that to every four-dimensional $\mathcal{N} = 2$ superconformal field theory one can associate a vertex operator algebra (VOA), leading to a canonical map,

$$\mathbb{V} : \{4d\,\mathcal{N} = 2\,\text{SCFTs}\} \longrightarrow \{\text{VOAs}\}\,. \qquad (2.16)$$

The correspondence encoded in this map enjoys a variety of remarkable features, many of which were uncovered in the original paper [33], but of which we will only mention a select few useful ones for our current purposes:

1. For local four-dimensional SCFTs, the corresponding vertex operator algebra has a Virasoro subalgebra with central charge $c_{2d}$ proportional to the four-dimensional $c$-anomaly coefficient: $c_{2d} = -12c_{4d}$.

2. Higgs branch chiral ring generators give rise to strong generators of the associated vertex operator algebra.[12] In particular, the moment map operators of the four-dimensional flavour symmetry map to affine Kac-Moody currents with affine level $k_{2d}$ determined in terms of the flavour central charge $k_{4d}$ according to $k_{2d} = -\frac{1}{2}k_{4d}$.

   In exceptional cases, the total Sugawara stress tensor constructed from the various affine currents provides the conformal vector of the VOA. A simple criterion for when this happens is whether the unitarity bound $c_{2d} \geqslant c_{\text{Sug,tot}}$ is saturated, where $c_{\text{Sug,tot}}$ is the total Sugawara central charge [33,47].[13]

3. As stated above, the vacuum character of the VOA is computed by the Schur limit of the superconformal index of the SCFT.[14,15] In particular, the two-dimensional conformal weight of the image of some operator $\mathcal{O}$ is given by $h_{\mathbb{V}(\mathcal{O})} = E_{\mathcal{O}} - R_{\mathcal{O}}$.

---

[11]The trace in this case runs over the full Hilbert space of states in radial quantisation, as in this limit pairwise cancellations automatically ensure that only states satisfying $E - (j_1 + j_2) - 2R = 0$ and $j_1 - j_2 + r = 0$ contribute.

[12]Strong generators are those operators in the VOA that cannot be written as normally ordered products of any other operators.

[13]The Sugawara central charge for the affinisation of a simple factor $\mathfrak{f}$ at level $k_{2d}$ is given by

$$c_{\text{Sug}} = \frac{k_{2d}\dim\mathfrak{f}}{k_{2d} + h_{\mathfrak{f}}^{\vee}}\,. \qquad (2.17)$$

For affine $\mathfrak{u}(1)$ factors (*i.e.*, Heisenberg vertex subalgebras) one has $c_{\text{Sug}} = 1$. The total Sugawara central charge is the sum of the contributions from all factors.

[14]This statement was proved using localisation techniques for Lagrangian theories in [48] (see also [49]). Similarly, using localisation techniques, one can attempt to carve out the full vertex operator algebra from the path integral, see, *e.g.*, [50]. Also, the VOA emerges by applying a suitable $\Omega$-deformation, see [51,52].

[15]The operator product algebraic structure of the VOA does not preserve the grading by the $SU(2)_R$ Cartan quantum number, although it does preserve its filtration [53]. Nevertheless, one can in principle refine the count of states of the vertex operator algebra to recover the Macdonald index. An early recipe to do so can be found in [54] (see also [55] for related work), and a universally applicable proposal based on free-field realisations was put forward in [31,56], see also [28].

4. The image of any SCFT under the correspondence $\mathbb{V}$ is independent of exactly marginal couplings. Thus VOAs are associated to whole conformal manifolds in the four- dimensional landscape.

Thanks to the fourth property, applying this map to theories of class $\mathcal{S}$ defines a TQFT on the surface $\mathcal{C}_{g,s}$ that takes values in vertex operator algebras [57,58]. The resulting class of VOAs have been called *chiral algebras of class* $\mathcal{S}$. The basic such chiral algebras are those associated with trinion theories. Exploratory studies and constructions of these VOAs were performed in [58,59], and a mathematical treatment of VOAs associated with untwisted trinions can be found in [60]. The vertex operator algebra captures a fairly intricate, infinite subset of the conformal data of an SCFT, and as such it is an indispensable structure in the study of four-dimensional superconformal field theories.

# 3   The twisted $A_2$ family

We now turn to our main objects of interest, the trinions for twisted $A_2$ theories. We will analyse in detail each of the five possible trinions and compute, using the general results from the previous section, $(i)$ their conformal anomaly coefficients $a$ and $c$, $(ii)$ their superconformal indices, which in particular provides us with access to detailed information on their number of free hypermultiplets and their possibly enhanced flavour symmetries, and $(iii)$ their associated vertex operator algebras. These data will prove sufficient to convincingly identify four out of the five theories with well-studied SCFTs, while the fifth has not yet appeared in the literature as far as the authors can tell. Surprisingly, our identifications include several Argyres-Douglas theories, *i.e.*, theories whose Coulomb branch chiral ring operators do not all have integer scaling dimensions.

To specify a twisted $A_2$ trinion theory $T_{\mathfrak{a}_2}^{\Lambda_1 \Lambda_2 \Lambda_3}$, we need to choose one embedding $\Lambda_1 : \mathfrak{su}(2) \hookrightarrow \mathfrak{a}_2$, as well as two embeddings $\Lambda_2, \Lambda_3 : \mathfrak{su}(2) \hookrightarrow \mathfrak{c}_1$. The options are quite limited and are summarised in Table 4.[16] All combinations of one untwisted and a pair of twisted punctures define good physical theories, except for the case where $\Lambda_1$ is subregular and $\Lambda_2 = \Lambda_3$ are principal, see also footnote 7. Table 1 provides an overview of the members of the twisted $A_2$ family and their properties; we will analyse each one of them in turn.

## 3.1   Theory 1: $\tilde{T}_3$

We start by considering the twisted $A_2$ theory whose three punctured are labelled by trivial embeddings $\Lambda_1 = 0 = \Lambda_{2,3}$. We call this theory $\tilde{T}_3$, *i.e.*,

$$\tilde{T}_3 \quad \longleftrightarrow \quad$$ 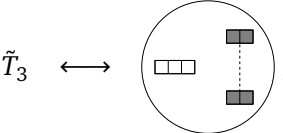

where we depict the $\mathfrak{su}(2)$ embeddings labelling the punctures by their corresponding Young diagram as in Table 4. As far as we know, the theory $\tilde{T}_3$ has not yet been investigated in the literature. We will analyse it in some detail.

---

[16]The principal $\mathfrak{su}(2)$ embedding into $\mathfrak{a}_2$ does not appear as it does not enter in a physical trinion theory. In fact, an untwisted puncture labelled by the principal embedding is simply absent. This is not the case for twisted punctures, where the principal embedding leaves behind a flavourless puncture that still carries the appropriate monodromy.

Table 4: All $\mathfrak{su}(2)$ embeddings that are relevant in defining twisted $A_2$ trinions. The first columns of subtables 4a and 4b give the names of the embeddings. The second columns provide Young diagrammatic depictions of the decomposition of the fundamental representation under the embedded $\mathfrak{su}(2)$: the heights of the columns of the Young diagrams encode the dimensions of the $\mathfrak{su}(2)$ representations that appear in the decompositions. The third columns give the commutant of the embedded $\mathfrak{su}(2)$, which is the manifest flavour symmetry associated with the puncture. The final columns provide the decompositions of the adjoint representation itself as in (2.1). Boldface numbers denote dimensions of representations.

| $\Lambda_1$ | $\mathbf{3}$ | $\mathfrak{f}_\Lambda$ | $\mathfrak{a}_2$ |
| --- | --- | --- | --- |
| trivial | ▭▭▭ | $\mathfrak{a}_2$ | $\mathfrak{a}_2$ |
| subregular | | $\mathfrak{u}_1$ | $\Lambda_1(\mathfrak{a}_1) \oplus \mathfrak{u}_1 \oplus \mathbf{2}_{+3} \oplus \mathbf{2}_{-3}$ |

(a) Possible embeddings $\Lambda_1 : \mathfrak{su}(2) \hookrightarrow \mathfrak{a}_2$.

| $\Lambda_{2,3}$ | $\mathbf{2}$ | $\mathfrak{f}_\Lambda$ | $\mathfrak{c}_1$ |
| --- | --- | --- | --- |
| trivial | ▦▦ | $\mathfrak{c}_1$ | $\mathfrak{c}_1$ |
| principal | | $\emptyset$ | $\Lambda_{2,3}(\mathfrak{a}_1)$ |

(b) Possible embeddings $\Lambda_{2,3} : \mathfrak{su}(2) \hookrightarrow \mathfrak{c}_1$.

### 3.1.1 Central charges, Macdonald index, and Higgs branch chiral ring

It is straightforward to compute the Weyl anomaly coefficients using the formulae (2.2),

$$a_{\tilde{T}_3} = \frac{11}{4}, \qquad c_{\tilde{T}_3} = 3.$$ 
(3.1)

The Shapere-Tachikawa relation [61], which is expected to hold for all the theories under investigation in this note, provides some insights into the Coulomb branch chiral ring of this theory. Indeed, the relation states that

$$4(2a_{\tilde{T}_3} - c_{\tilde{T}_3}) = 10 = \sum_i (2\Delta_i - 1),$$ 
(3.2)

where the sum runs over the generators of the Coulomb branch chiral ring and $\Delta_i$ represents their scaling dimensions/$U(1)_r$ charges. One immediately observes that this theory cannot be of rank one, *i.e.*, its Coulomb branch cannot have complex dimension one or, equivalently, its chiral ring cannot be generated by a single generator, as $\Delta = \frac{11}{2}$ is not an allowed value for a scaling dimension at rank one [62]. We will argue below (3.29) that this theory has rank two, with $\Delta_1 = \Delta_2 = 3$.

The Macdonald index for this theory can be evaluated using the expression (2.9). One finds[17]

$$
\begin{aligned}
I_M^{\tilde{T}_3}(q,t;\mathbf{a},\mathbf{b}_j) &= \sum_\lambda \frac{\mathrm{PE}\big[\frac{t}{1-q}\chi_{\mathrm{adj}}^{\mathfrak{a}_2}(\mathbf{a})\big] P_{(\lambda,\lambda)}^{\mathfrak{a}_2}(\mathbf{a}) \prod_{j=1}^2 \mathrm{PE}\big[\frac{t}{1-q}\chi_{\mathrm{adj}}^{\mathfrak{c}_1}(\mathbf{b}_j)\big] P_{(\lambda)}^{\mathfrak{c}_1}(\mathbf{b}_j)}{\mathrm{PE}\big[\frac{t^2+t^3}{1-q}\big] P_{(\lambda,\lambda)}^{\mathfrak{a}_2}(t,1,t^{-1})} \\
&= \mathrm{PE}\Big[\frac{1}{1-q}\Big\{\big(\chi_{\mathrm{adj}}^{\mathfrak{c}_1}(\mathbf{b}_1) + \chi_{\mathrm{adj}}^{\mathfrak{c}_1}(\mathbf{b}_2) + \chi_{\mathrm{adj}}^{\mathfrak{a}_2}(\mathbf{a})\big)t + \big(\chi_{\mathbf{2}}^{\mathfrak{c}_1}(\mathbf{b}_1)\chi_{\mathbf{2}}^{\mathfrak{c}_1}(\mathbf{b}_2)\chi_{\mathrm{adj}}^{\mathfrak{a}_2}(\mathbf{a}) - 2\big)t^2 + qt \\
&\quad + \chi_{\mathbf{2}}^{\mathfrak{a}_1}(\mathbf{b}_1)\chi_{\mathbf{2}}^{\mathfrak{a}_1}(\mathbf{b}_2)qt^2 - \big(\chi_{\mathbf{2}}^{\mathfrak{a}_1}(\mathbf{b}_1)\chi_{\mathbf{2}}^{\mathfrak{a}_1}(\mathbf{b}_2) + 3\chi_{\mathbf{2}}^{\mathfrak{a}_1}(\mathbf{b}_1)\chi_{\mathbf{2}}^{\mathfrak{a}_1}(\mathbf{b}_2)\chi_{\mathrm{adj}}^{\mathfrak{a}_2}(\mathbf{a}) + 1\big)t^3 + \dots\Big\}\Big].
\end{aligned}
$$ 
(3.3)

In the first line, the sum runs over all Dynkin labels $\lambda$ of $\mathfrak{c}_1$ representations, *i.e.*, all positive integers, and we used Table 3 to assign the $\mathfrak{a}_2$ representation with Dynkin labels $(\lambda,\lambda)$ to each

---

[17]Note that under the principal embedding $\rho : \mathfrak{su}(2) \hookrightarrow \mathfrak{su}(3)$ one has $\rho(\sigma_3) = \left(\begin{smallmatrix} 2 & 0 & 0 \\ 0 & 0 & 0 \\ 0 & 0 & -2 \end{smallmatrix}\right)$. Hence $t^{\frac{1}{2}\rho(\sigma_3)}$ has diagonal entries $t, 1, t^{-1}$, which we have indicated in the argument of the Macdonald polynomial in the denominator. This is slightly redundant information, as their product is naturally constrained to be one.

$\mathfrak{c}_1$ representation with Dynkin label $(\lambda)$.[18] We have expressed the result in terms of a plethystic exponential, which facilitates an analysis of generators and relations. Indeed, we immediately see that no free hypermultiplets are present and that the flavour symmetry of the theory is not enhanced beyond the manifest symmetry captured by the punctures. These punctures are all full, thus their flavour central charges (2.7) equal twice the respective dual Coxeter numbers,

$$G_F^{\tilde{T}_3} = SU(2)_4^{(1)} \times SU(2)_4^{(2)} \times SU(3)_6 \,. \tag{3.4}$$

Each of the $SU(2)$ factors carries a Witten anomaly. More importantly, we have access to a wealth of information about the Higgs branch chiral ring generators and their relations, and about any additional strong VOA generators.

Taking the $q \to 0$ limit of (3.3) returns the Hilbert series of the Higgs branch of $\tilde{T}_3$. Analysing this expression shows that the moment map operators of the flavour symmetry $G_F^{\tilde{T}_3}$ are, as expected, among the generators of the Higgs branch chiral ring of $\tilde{T}_3$. These operators have $E = 2R = 2$ and transform in the adjoint representation of the respective flavour symmetry factors. We will denote them as $\mu_{\mathfrak{su}(2)}^{(1)}, \mu_{\mathfrak{su}(2)}^{(2)}, \mu_{\mathfrak{su}(3)}$. Moreover, there is an additional generator with $E = 2R = 4$, which transforms in the representation $(\mathbf{2}, \mathbf{2}, \mathbf{8})$ of $G_F^{\tilde{T}_3}$. We will denote this generator by $\omega$. These generators satisfy a number of elementary relations, the quantum numbers of which can be read off from the index. As explained in [53], the explicit expressions for these relations can be deduced from null relations in the associated vertex operator algebra, which we present below.[19] The relations—organised by their $SU(2)_R$ charges—are as follows,

$$R = 2: \qquad \left(\mu_{\mathfrak{su}(2)}^{(1)}\right)^2\Big|_{(\mathbf{1},\mathbf{1},\mathbf{1})} = \left(\mu_{\mathfrak{su}(2)}^{(2)}\right)^2\Big|_{(\mathbf{1},\mathbf{1},\mathbf{1})} = \frac{1}{4}\mu_{\mathfrak{su}(3)}^2\Big|_{(\mathbf{1},\mathbf{1},\mathbf{1})} \,, \tag{3.5}$$

$$R = 3: \qquad \mu_{\mathfrak{su}(3)}^3\Big|_{(\mathbf{1},\mathbf{1},\mathbf{1})} = 0 \,, \tag{3.6}$$

$$\mu_{\mathfrak{su}(3)}\,\omega\Big|_{(\mathbf{2},\mathbf{2},\mathbf{1})} = 0 \,, \tag{3.7}$$

$$\mu_{\mathfrak{su}(3)}\,\omega\Big|_{(\mathbf{2},\mathbf{2},\mathbf{8}_s)} = 0 \,, \tag{3.8}$$

$$\mu_{\mathfrak{su}(3)}\,\omega\Big|_{(\mathbf{2},\mathbf{2},\mathbf{8}_a)} = 4\,\mu_{\mathfrak{su}(2)}^{(1)}\,\omega\Big|_{(\mathbf{2},\mathbf{2},\mathbf{8})} = 4\,\mu_{\mathfrak{su}(2)}^{(2)}\,\omega\Big|_{(\mathbf{2},\mathbf{2},\mathbf{8})} \,. \tag{3.9}$$

Here the subscript $s$ or $a$ on $\mathbf{8}$ indicates whether the representation appears in the symmetric or the antisymmetric tensor product of two adjoint representations of $\mathfrak{su}(3)$, respectively. One encounters further relations at $R = 4$, relating the square of $\omega$ and quartic expressions of moment maps, transforming in the following representations,[20]

$$R = 4: \qquad (\mathbf{1}, \mathbf{1}, \mathbf{27} \oplus \mathbf{8}_s \oplus \mathbf{1}) \,, \tag{3.10}$$

$$(\mathbf{1}, \mathbf{3}, \mathbf{8}_a \oplus \mathbf{10} \oplus \overline{\mathbf{10}}) \oplus (\mathbf{3}, \mathbf{1}, \mathbf{8}_a \oplus \mathbf{10} \oplus \overline{\mathbf{10}}) \,, \tag{3.11}$$

$$(\mathbf{3}, \mathbf{3}, \mathbf{8}_s \oplus \mathbf{1}) \,. \tag{3.12}$$

For example, $\omega^2\big|_{(\mathbf{3},\mathbf{3},\mathbf{8}_s\oplus\mathbf{1})} \sim \mu_{\mathfrak{su}(2)}^{(1)}\mu_{\mathfrak{su}(2)}^{(2)}\mu_{\mathfrak{su}(3)}^2\big|_{(\mathbf{3},\mathbf{3},\mathbf{8}_s\oplus\mathbf{1})}$. We suspect this is the full list of elementary relations of the Higgs branch chiral ring of $\tilde{T}_3$ We note that the relations (3.5)–(3.6) are canonical Higgs branch relation in theories of class $\mathcal{S}$ (see, *e.g.*, [27]), with the first being a direct consequence of the "criticality" of the flavour central charges $k = 2h^\vee$ [47].

---

[18]We use interchangeably Dynkin labels and boldfaced dimensions to denote representations. In particular, we have $(1) = \mathbf{2}$ and $(2) = \mathbf{3} = \mathrm{adj}$ for $\mathfrak{c}_1$, and $(1,0) = \mathbf{3}$ and $(1,1) = \mathbf{8} = \mathrm{adj}$ for $\mathfrak{a}_2$.

[19]The procedure amounts to setting to zero all composites operators containing derivatives as well as any operators that are nilpotent up to composites containing derivatives. This is equivalent to passing to the reduced version of Zhu's $C_2$ algebra of the VOA.

[20]Note that $\mathbf{27} = (2,2)$ is the $\mathfrak{su}(3)$ representation with Dynkin labels twice those of the adjoint representation. Furthermore, $\mathbf{10} = (3,0)$ and $\overline{\mathbf{10}} = (0,3)$.

### 3.1.2 The $\tilde{T}_3$ vertex operator algebra

The Higgs branch chiral ring generators $\mu_{\mathfrak{su}(2)}^{(1)}, \mu_{\mathfrak{su}(2)}^{(2)}, \mu_{\mathfrak{su}(3)}$ and $\omega$ give rise to strong generators of the vertex operator algebra $\mathbb{V}(\tilde{T}_3)$. In particular, the moment maps turn into affine currents of critical level $k_{2d} = -h^\vee$. The Macdonald index indicates the presence of one additional strong generator, namely the stress energy tensor, which descends from a component of the $SU(2)_R$ conserved current multiplet and is captured by the $+qt$ term in the plethystic logarithm of the index.[21] The Virasoro central charge is easily ascertained: $c_{2d} = -12c_{\tilde{T}_3} = -36$. What's more, the Macdonald index indicates that the relation (3.7) is not realised as a null relation in the VOA, but instead leads to an operator of lower $R$-grading than expected in the $R$-filtration of [53], see also [28, 31]. Table 5 summarises our list of strong generators for $\mathbb{V}(\tilde{T}_3)$.

We now turn to the task of constructing the vertex operator algebra $\mathbb{V}(\tilde{T}_3)$. We do so by bootstrapping the singular terms in the operator product expansions (OPEs) of the strong generators presented in Table 5. In practice, this means we formulate the most general expressions for these singular OPEs that are compatible with the global symmetries in terms of a number of undetermined numerical coefficients, and we then demand that the Jacobi identities hold modulo null states. These constraints result in a set of algebraic equations for the numerical coefficients that can be solved if the set of strong generators describes a consistent VOA.[22] We take a successful solution to be an indication that the VOA is correctly identified, although strictly speaking we could be producing only a vertex operator subalgebra of the full VOA.

The Virasoro and affine current subalgebras take their canonical forms,

$$T(z)\, T(0) \sim \frac{-18}{z^4} + \frac{2T}{z^2} + \frac{\partial T}{z}\,, \tag{3.13}$$

$$j_{ij}^{(1)}(z)\, j_{kl}^{(1)}(0) \sim \frac{-2\,\epsilon_{l(i}\epsilon_{j)k}}{z^2} + \frac{2\epsilon_{(i(k}\, j_{j)l)}^{(1)}(0)}{z}\,, \tag{3.14}$$

$$j_{\alpha\beta}^{(2)}(z)\, j_{\gamma\delta}^{(2)}(0) \sim \frac{-2\,\epsilon_{\delta(\alpha}\epsilon_{\beta)\gamma}}{z^2} + \frac{2\epsilon_{(\alpha(\gamma}\, j_{\beta)\delta)}^{(2)}(0)}{z}\,, \tag{3.15}$$

$$J_{b_1}^{a_1}(z)\, J_{b_2}^{a_2}(0) \sim \frac{-3\,\Delta_{b_1 b_2}^{a_1 a_2}}{z^2} + \frac{if_{b_1 b_2 b}^{a_1 a_2 a}\, J_a^b(0)}{z}\,. \tag{3.16}$$

We have introduced realisations of the Killing form and structure constants of $\mathfrak{su}(3)$ in terms of our index conventions,

$$\Delta_{b_1 b_2}^{a_1 a_2} := \delta_{b_2}^{a_1}\delta_{b_1}^{a_2} - \frac{1}{3}\delta_{b_1}^{a_1}\delta_{b_2}^{a_2}\,, \qquad f^{a_1 a_2 a_3} := -i\big(\delta_{b_2}^{a_1}\delta_{b_3}^{a_2}\delta_{b_1}^{a_3} - \delta_{b_3}^{a_1}\delta_{b_1}^{a_2}\delta_{b_2}^{a_3}\big)\,. \tag{3.17}$$

The OPE of the stress tensor with the other strong generators also takes a canonical form

$$T(z)\, \mathcal{V}(0) \sim \frac{h_\mathcal{V}\, \mathcal{V}(0)}{z^2} + \frac{\partial\mathcal{V}(0)}{z}\,, \tag{3.18}$$

while the affine currents from different simple factors of the flavour symmetry mutually commute and their operator product expansions with the primary $W$ are determined by its trans-

---

[21] Indeed, such a stress tensor generator must be present unless the stress tensor is the Sugawara vector of the theory's affine symmetry, and in this case all affine subalgebras are at their critical levels, so cannot provide a normalised stress tensor.

[22] Here and in the remainder of the paper, the computational strategy for bootstrapping VOAs has been implemented in Mathematica using the package OPEdefs [63].

Table 5: Strong generators of the vertex operator algebra associated with $\tilde{T}_3$. Our index convention is that the adjoint representation of $\mathfrak{su}(2)$ is realised by a symmetrised pair of fundamental indices and the adjoint representation of $\mathfrak{su}(3)$ is realised by the traceless combination of a fundamental and antifundamental index. Fundamental indices of $\mathfrak{su}(2)^{(1)}$ are denoted as $\{i, j, k, \ldots\}$, while those of $\mathfrak{su}(2)^{(2)}$ are denoted as $\{\alpha, \beta, \gamma, \ldots\}$ and fundamental or anti-fundamental indices of $\mathfrak{su}(3)$ are denoted as lowered or raised $\{a, b, c, \ldots\}$.

| $\mathcal{O}$ | $\mathbb{V}(\mathcal{O})$ | $h_{\mathbb{V}(\mathcal{O})}$ | $\mathfrak{su}(2)^{(1)} \times \mathfrak{su}(2)^{(2)} \times \mathfrak{su}(3)$ |
|---|---|---|---|
| $\mu_{\mathfrak{su}(2)}^{(1)}$ | $j_{ij}^{(1)}$ | 1 | $(\mathbf{3}, \mathbf{1}, \mathbf{1})$ |
| $\mu_{\mathfrak{su}(2)}^{(2)}$ | $j_{\alpha\beta}^{(2)}$ | 1 | $(\mathbf{1}, \mathbf{3}, \mathbf{1})$ |
| $\mu_{\mathfrak{su}(3)}$ | $J_b^a$ | 1 | $(\mathbf{1}, \mathbf{1}, \mathbf{8})$ |
| $\omega$ | $W_{i\alpha b}{}^a$ | 2 | $(\mathbf{2}, \mathbf{2}, \mathbf{8})$ |
| $J_{+\dot{+}}^{11}$ | $T$ | 2 | $(\mathbf{1}, \mathbf{1}, \mathbf{1})$ |

formation properties under the global $\mathfrak{su}(2)^{(1)} \times \mathfrak{su}(2)^{(2)} \times \mathfrak{su}(3)$,

$$j_{ij}^{(1)}(z)\, W_{k\alpha b}{}^a(0) \sim \frac{-\frac{1}{2}\epsilon_{k(i}\, W_{j)\alpha b}{}^a(0)}{z}\,, \tag{3.19}$$

$$j_{\alpha\beta}^{(2)}(z)\, W_{i\gamma b}{}^a(0) \sim \frac{-\frac{1}{2}\epsilon_{\gamma(\alpha}\, W_{i\beta)b}{}^a(0)}{z}\,, \tag{3.20}$$

$$J_{b_1}^{a_1}(z)\, W_{i\alpha b_2}{}^{a_2}(0) \sim \frac{if_{b_1 b_2 b}^{a_1 a_2 a}\, W_{i\alpha a}{}^b(0)}{z}\,. \tag{3.21}$$

Before presenting the $W \times W$ OPE, it will be helpful to introduce one additional tensor,

$$d_{b_1 b_2 b_3}^{a_1 a_2 a_3} := \delta_{b_2}^{a_1}\delta_{b_3}^{a_2}\delta_{b_1}^{a_3} + \delta_{b_3}^{a_1}\delta_{b_1}^{a_2}\delta_{b_2}^{a_3} - \frac{2}{3}\delta_{b_1}^{a_1}\delta_{b_3}^{a_2}\delta_{b_2}^{a_3} - \frac{2}{3}\delta_{b_2}^{a_2}\delta_{b_3}^{a_1}\delta_{b_1}^{a_3} - \frac{2}{3}\delta_{b_3}^{a_3}\delta_{b_2}^{a_1}\delta_{b_1}^{a_2} + \frac{4}{9}\delta_{b_1}^{a_1}\delta_{b_2}^{a_2}\delta_{b_3}^{a_3}\,, \tag{3.22}$$

which is the cubic Casimir expressed in terms of (anti)fundamental indices. The $W \times W$ OPE now reads[23]

$$W_{i\alpha b_1}{}^{a_1}(z)\, W_{j\beta b_2}{}^{a_2}(0) \sim \frac{\epsilon_{ij}\,\epsilon_{\alpha\beta}\,\Delta_{b_1 b_2}^{a_1 a_2}}{z^4} + \frac{1}{z^3}\Big(\frac{1}{2}\big(\epsilon_{\alpha\beta}\, j_{ij}^{(1)} + \epsilon_{ij}\, j_{\alpha\beta}^{(2)}\big)\Delta_{b_1 b_2}^{a_1 a_2} - \frac{1}{3}\epsilon_{ij}\,\epsilon_{\alpha\beta}\, if_{b_1 b_2 b}^{a_1 a_2 a}\, J_a^b\Big)$$

$$+ \frac{1}{z^2}\Big(\frac{1}{4}\big(\epsilon_{\alpha\beta}\, \partial j_{ij}^{(1)} + \epsilon_{ij}\, \partial j_{\alpha\beta}^{(2)}\big)\Delta_{b_1 b_2}^{a_1 a_2} - \frac{1}{6}\epsilon_{ij}\,\epsilon_{\alpha\beta}\, if_{b_1 b_2 b}^{a_1 a_2 a}\, \partial J_a^b - \frac{1}{4}\epsilon_{ij}\,\epsilon_{\alpha\beta}\, \Delta_{b_1 b_2}^{a_1 a_2}\, T$$

$$+ \frac{11}{96}\epsilon_{ij}\,\epsilon_{\alpha\beta}\, \Delta_{b_1 b_2}^{a_1 a_2}\, (J_b^a J_a^b) + \frac{1}{8} d_{b_1 b_2 b}^{a_1 a_2 a}\, d_{b_4 b_5 a}^{a_4 a_5 b}\, (J_{a_4}^{b_4} J_{a_5}^{b_5}) - \frac{1}{24}\epsilon_{ij}\,\epsilon_{\alpha\beta}\, \big((J_{b_1}^{a_1} J_{b_2}^{a_2}) + (J_{b_2}^{a_2} J_{b_1}^{a_1})\big)$$

$$+ \frac{1}{4}\Delta_{b_1 b_2}^{a_1 a_2}\, (j_{ij}^{(1)} j_{\alpha\beta}^{(2)}) + \frac{1}{6} if_{b_1 b_2 b}^{a_1 a_2 a}\, \big(\epsilon_{\alpha\beta}\, (j_{ij}^{(1)} J_a^b) + \epsilon_{ij}\, (j_{\alpha\beta}^{(2)} J_a^b)\big) + \frac{3}{8}\epsilon_{ij}\,\epsilon_{\alpha\beta}\, d_{b_1 b_2 b}^{a_1 a_2 a}\, \partial J_a^b\Big)$$

---

[23]All structures on the right-hand side have definite representation-theoretic properties under $\mathfrak{su}(2)^{(1)} \times \mathfrak{su}(2)^{(2)} \times \mathfrak{su}(3)$, with two exceptions. Instead of constructing the representation $(2, 2) = \mathbf{27}$ in the symmetrised product of two $\mathfrak{su}(3)$ adjoint objects, we have used the symmetrised product itself. To obtain the "pure" $(2, 2)$ contribution, one should subtract the contribution of the other two representations that appear in that product, i.e., $(1, 1) = \mathbf{8}_s$ and $(0, 0) = \mathbf{1}$. Similarly, instead of constructing $(3, 0) \oplus (0, 3)$ in the antisymmetrised product of two $\mathfrak{su}(3)$ adjoint objects, we have used the antisymmetrised product itself, from which the contribution of $\mathbf{8}_a$ can be subtracted.

$$
\begin{aligned}
+\frac{1}{z}\Big(&\frac{1}{16}\big(\epsilon_{\alpha\beta}\,\partial^2 j^{(1)}_{ij}+\epsilon_{ij}\,\partial^2 j^{(2)}_{\alpha\beta}\big)\Delta^{a_1 a_2}_{b_1 b_2}-\frac{1}{24}\epsilon_{ij}\,\epsilon_{\alpha\beta}\,if^{a_1 a_2 a}_{b_1 b_2 b}\,\partial^2 J^b_a-\frac{1}{8}\epsilon_{ij}\,\epsilon_{\alpha\beta}\,\Delta^{a_1 a_2}_{b_1 b_2}\,\partial T\\
&-\frac{1}{8}\Delta^{a_1 a_2}_{b_1 b_2}\big(\epsilon_{\alpha\beta}\,\epsilon^{kl}\,(j^{(1)}_{k(i}\partial j^{(1)}_{j)l})+\epsilon_{ij}\,\epsilon^{\gamma\delta}\,(j^{(2)}_{\gamma(\alpha}\partial j^{(2)}_{\beta)\delta})\big)+\frac{11}{96}\epsilon_{ij}\,\epsilon_{\alpha\beta}\,\Delta^{a_1 a_2}_{b_1 b_2}\,(J^a_b\partial J^b_a)\\
&+\frac{1}{8}d^{a_1 a_2 b}_{b_1 b_2 b}\,d^{a_4 a_5 b}_{b_4 b_5 a}\,(J^{b_4}_{a_4}\partial J^{b_5}_{a_5})-\frac{1}{24}\epsilon_{ij}\,\epsilon_{\alpha\beta}\big((J^{a_1}_{b_1}\partial J^{a_2}_{b_2})+(J^{a_2}_{b_2}\partial J^{a_1}_{b_1})\big)\\
&-\frac{1}{12}\epsilon_{ij}\,\epsilon_{\alpha\beta}\big((J^{a_1}_{b_1}\partial J^{a_2}_{b_2})-(J^{a_2}_{b_2}\partial J^{a_1}_{b_1})\big)+\frac{1}{8}\Delta^{a_1 a_2}_{b_1 b_2}\big((j^{(1)}_{ij}\partial j^{(2)}_{\alpha\beta})+(\partial j^{(1)}_{ij}j^{(2)}_{\alpha\beta})\big)\\
&+\big(\tfrac{1}{12}if^{a_1 a_2 a}_{b_1 b_2 b}+\tfrac{3}{16}d^{a_1 a_2 a}_{b_1 b_2 b}\big)\big(\epsilon_{\alpha\beta}\,(j^{(1)}_{ij}\partial J^b_a)+\epsilon_{ij}\,(j^{(2)}_{\alpha\beta}\partial J^b_a)\big)\\
&+\frac{1}{12}if^{a_1 a_2 a}_{b_1 b_2 b}\big(\epsilon_{\alpha\beta}\,(\partial j^{(1)}_{ij}J^b_a)+\epsilon_{ij}\,(\partial j^{(2)}_{\alpha\beta}J^b_a)\big)-\frac{1}{8}\big(\epsilon_{\alpha\beta}\,(Tj^{(1)}_{ij})+\epsilon_{ij}\,(Tj^{(2)}_{\alpha\beta})\big)\Delta^{a_1 a_2}_{b_1 b_2}\\
&-\frac{1}{12}\epsilon_{ij}\,\epsilon_{\alpha\beta}\,if^{a_1 a_2 a}_{b_1 b_2 b}\,(TJ^b_a)+\frac{1}{12}if^{a_1 a_2 a}_{b_1 b_2 b}\,(j^{(1)}_{ij}j^{(2)}_{\alpha\beta}J^b_a)+\frac{1}{24}\Delta^{a_1 a_2}_{b_1 b_2}\big(\epsilon_{\alpha\beta}(j^{(1)}_{ij}J^a_b J^b_a)+\epsilon_{ij}(j^{(2)}_{\alpha\beta}J^a_b J^b_a)\big)\\
&+\frac{1}{16}d^{a_1 a_2 a}_{b_1 b_2 b}\,d^{a_4 a_5 b}_{b_4 b_5 a}\big(\epsilon_{\alpha\beta}\,(j^{(1)}_{ij}J^{b_4}_{a_4}J^{b_5}_{a_5})+\epsilon_{ij}\,(j^{(2)}_{\alpha\beta}J^{b_4}_{a_4}J^{b_5}_{a_5})\big)+\frac{7}{288}\epsilon_{ij}\,\epsilon_{\alpha\beta}\,if^{a_1 a_2 a}_{b_1 b_2 b}\,(J^b_a J^{a'}_{b'}J^{b'}_{a'})\\
&-\frac{1}{48}\big(\epsilon_{\alpha\beta}\,(j^{(1)}_{ij}(J^{a_1}_{b_1}J^{a_2}_{b_2}+J^{a_2}_{b_2}J^{a_1}_{b_1}))+\epsilon_{ij}\,(j^{(2)}_{\alpha\beta}(J^{a_1}_{b_1}J^{a_2}_{b_2}+J^{a_2}_{b_2}J^{a_1}_{b_1}))\big)\\
&-\frac{1}{24}\epsilon_{ij}\,\epsilon_{\alpha\beta}\big(\varepsilon_{b_1 b_2 c}\,\varepsilon^{ac(d}\,(J^{a_1}_d J^{a_2)}_b J^b_a)-\varepsilon^{a_1 a_2 c}\,\varepsilon_{ac(d}\,(J^d_{b_1}J^b_{b_2)}J^a_b)\big)\Big)\,.
\end{aligned}
\tag{3.23}
$$

In this expression, we have omitted the positional argument of the operators on the right-hand side—they are all inserted at the origin. There are also null operators that can be added freely on the right-hand side. In particular, we have made the choice to include the term proportional to $(J^a_b J^b_a)$ and no terms proportional to $\epsilon^{ik}\,\epsilon^{jl}\,(j^{(1)}_{ij}j^{(1)}_{kl})$ or $\epsilon^{\alpha\gamma}\,\epsilon^{\beta\delta}\,(j^{(2)}_{\alpha\beta}j^{(2)}_{\gamma\delta})$, as these are interchangeable due to null relations associated with the Higgs branch relations (3.5). Indeed, the explicit null relations at conformal weight two are

$$
\frac{1}{4}(J^a_b J^b_a)=\epsilon^{il}\,\epsilon^{kj}\,(j^{(1)}_{ij}j^{(1)}_{kl})=\epsilon^{\alpha\delta}\,\epsilon^{\gamma\beta}\,(j^{(2)}_{\alpha\beta}j^{(2)}_{\gamma\delta})\,.
\tag{3.24}
$$

At weight three, one finds in the representation $(\mathbf{1},\mathbf{1},\mathbf{1})$ the null fields

$$
(J^a_b J^b_c J^c_a)=-\frac{3}{4}\partial(J^d_e J^e_d)=-3\epsilon^{il}\,\epsilon^{kj}\,\partial(j^{(1)}_{ij}j^{(1)}_{kl})=-3\epsilon^{\alpha\delta}\,\epsilon^{\gamma\beta}\,\partial(j^{(2)}_{\alpha\beta}j^{(2)}_{\gamma\delta})\,,
\tag{3.25}
$$

while in representation $(\mathbf{2},\mathbf{2},\mathbf{8})$, one has

$$
0=(J^c_b W_{i\alpha c}{}^a)+(J^a_d W_{i\alpha b}{}^d)-\frac{2}{3}\delta^a_b(J^c_d W_{i\alpha c}{}^d)\,,
\tag{3.26}
$$

and

$$
\epsilon^{kl}\,(j^{(1)}_{ik}W_{l\alpha b}{}^a)=\epsilon^{\gamma\delta}\,(j^{(2)}_{\alpha\gamma}W_{i\delta b}{}^a)=\frac{1}{4}\big((J^a_c W_{i\alpha b}{}^c)-(J^c_b W_{i\alpha c}{}^a)\big)\,.
\tag{3.27}
$$

Recall also that the Higgs branch relation at $R=3$ in the representation $(\mathbf{2},\mathbf{2},\mathbf{1})$ is not realised as a null relation.

## 3.2 Theory 2: rank-one $C_2 U_1$ theory

We next turn to the twisted $A_2$ trinion for which the untwisted puncture is specified by the subregular embedding $\Lambda_1:\mathfrak{su}(2)\to\mathfrak{su}(3)$, while the twisted punctures are still full punctures. In [8], exploiting an $S$-duality argument, this theory was identified as one of the "new rank-one SCFTs" discovered in [17, 18]. Some of its properties were further analysed in [19], and ultimately this theory was incorporated into the classification of [64]—it is the $IV^*$ theory in the $I_4$ series. We will refer to it as the rank-one $C_2 U_1$ theory, in reference to its $C_2\times U(1)$

flavour symmetry,

$$\text{rank-one } C_2U_1 \text{ theory} \quad \longleftrightarrow \quad \tag{3.28}$$

Let us expand on the analysis already available in the literature by, in particular, showcasing the full set of relations of its Higgs branch chiral ring and constructing its associated vertex operator algebra.

### 3.2.1 Central charges, Macdonald index, and Higgs branch chiral ring

The Weyl anomaly coefficients of the rank-one $C_2U_1$ theory are well-known and can be easily verified using (2.2),

$$a_{C_2U_1} = \frac{17}{12}\,, \qquad c_{C_2U_1} = \frac{19}{12}\,. \tag{3.29}$$

The unique Coulomb branch chiral ring generator for this theory has scaling dimension $\Delta = 3$, in agreement with the Shapere-Tachikawa relation. Knowledge of the Coulomb branch spectrum of this theory immediately grants access to the Coulomb branch spectrum of the $\tilde{T}_3$ theory of the previous subsection. Indeed, while remaining agnostic regarding the local contributions of twisted punctures to the Coulomb branch dimensions, it is known (see, e.g., [6]) that modifying an untwisted $A_2$ puncture from subregular to trivial has the effect of introducing one new Coulomb branch operator of dimension three. We thus see that $\tilde{T}_3$ has rank two and Coulomb branch scaling dimensions $\Delta_1 = \Delta_2 = 3$, as advertised above.

The Macdonald limit of the superconformal index of the $C_2U_1$ theory is straightforward to compute using (2.9)

$$
\begin{aligned}
I_M^{C_2U_1}(q,t;a,\mathbf{b}) &= \sum_\lambda \frac{\text{PE}\!\left[\frac{t+t^2+(a^3+a^{-3})t^{\frac{3}{2}}}{1-q}\right] P^{\mathfrak{a}_2}_{(\lambda,\lambda)}\!\left(a\sqrt{t},\frac{a}{\sqrt{t}},a^{-2}\right) \prod_{j=1}^2 \text{PE}\!\left[\frac{t}{1-q}\chi^{\mathfrak{c}_1}_{\text{adj}}(\mathbf{b}_j)\right] P^{\mathfrak{c}_1}_{(\lambda)}(\mathbf{b}_j)}{\text{PE}\!\left[\frac{t^2+t^3}{1-q}\right] P^{\mathfrak{a}_2}_{(\lambda,\lambda)}(t,1,t^{-1})} \\
&= \text{PE}\Big[\frac{1}{1-q}\Big\{\big(1+\chi^{\mathfrak{c}_2}_{\text{adj}}(\mathbf{b})\big)t + \big((a^3+a^{-3})\chi^{\mathfrak{c}_2}_{\mathbf{5}}(\mathbf{b})\big)t^{\frac{3}{2}} - \big(1+\chi^{\mathfrak{c}_2}_{\mathbf{5}}(\mathbf{b})\big)t^2 \\
&\quad + qt - \big((a^3+a^{-3})\big(\chi^{\mathfrak{c}_2}_{\text{adj}}(\mathbf{b})+\chi^{\mathfrak{c}_2}_{\mathbf{5}}(\mathbf{b})\big)\big)t^{\frac{5}{2}} \\
&\quad + \big(-(a^6+a^{-6})-1-\chi^{\mathfrak{c}_2}_{\mathbf{14}}(\mathbf{b})-\chi^{\mathfrak{c}_2}_{\text{adj}}(\mathbf{b})+\chi^{\mathfrak{c}_2}_{\mathbf{5}}(\mathbf{b})\big)t^3+\dots\Big\}\Big]\,.
\end{aligned}
\tag{3.30}
$$

We observe that the trinion does not contain free hypermultiplets and that the manifest $U(1) \times USp(2)_4 \times USp(2)_4$ flavour symmetry is indeed enhanced to $U(1) \times USp(4)_4$. The level of the latter is determined by the levels of the $USp(2)$ factors which are embedded in $USp(4)$ with embedding index one. We have written the result in terms of characters of this enhanced flavour symmetry. The $USp(2)_4$ factors each carry a Witten anomaly, and so does their enhancement $USp(4)_4$ [20].

From the $q \to 0$ limit of (3.30) we deduce that the Higgs branch chiral ring is generated by moment map operators $\mu_{\mathfrak{u}(1)}$ and $\mu_{\mathfrak{c}_2}$ of $R$-charge $R = 1$ and by $R = 3/2$ generators transforming as $\mathbf{5}_{\pm 3}$, which we denote $w_\pm$. The quantum numbers of their relations can also be discerned from (3.30). The explicit expressions of the elementary relations up to $R = 3$, as obtained from null operators in the vertex operator algebra to appear below, are

$$R = 2: \qquad \mu^2_{\mathfrak{c}_2}\big|_{\mathbf{1}_0} = -\frac{1}{9}\mu^2_{\mathfrak{u}(1)}\big|_{\mathbf{1}_0}\,, \tag{3.31}$$

$$\mu^2_{\mathfrak{c}_2}\big|_{\mathbf{5}_0} = 0\,, \tag{3.32}$$

$$R = \tfrac{5}{2}: \qquad \mu_{\mathfrak{c}_2} w_{\pm}\big|_{\mathbf{10}_{\pm 3}} = 0 , \tag{3.33}$$

$$\mu_{\mathfrak{c}_2} w_{\pm}\big|_{\mathbf{5}_{\pm 3}} = -\frac{1}{3}\mu_{\mathfrak{u}(1)} w_{\pm}\big|_{\mathbf{5}_{\pm 3}} , \tag{3.34}$$

$$R = 3: \qquad w_{\pm} w_{\pm}\big|_{\mathbf{1}_{\pm 6}} = 0 , \tag{3.35}$$

$$w_+ w_-\big|_{\mathbf{1}_0} = -\frac{2}{81}\mu_{\mathfrak{u}(1)}^3\big|_{\mathbf{1}_0} , \tag{3.36}$$

$$w_+ w_-\big|_{\mathbf{10}_0} = -\frac{1}{27}\mu_{\mathfrak{u}(1)}^2\mu_{\mathfrak{c}_2}\big|_{\mathbf{10}_0} , \tag{3.37}$$

$$w_+ w_-\big|_{\mathbf{14}_0} = -\frac{2}{9}\mu_{\mathfrak{u}(1)}\mu_{\mathfrak{c}_2}^2\big|_{\mathbf{14}_0} . \tag{3.38}$$

The term $+\chi_{\mathbf{5}}^{\mathfrak{c}_2}(\mathbf{b})t^3$ in the plethystic logarithm of $I_M^{C_2 U_1}(q,t;a,\mathbf{b})$ is an artefact of that logarithm and does not indicate the presence of an additional generator. Indeed, the plethystic exponential is designed to subtract all composites of the generators and the elementary relations. In particular it subtracts the composites of $\mu_{\mathfrak{c}_2}$ and the relation $\mu_{\mathfrak{c}_2}^2\big|_{\mathbf{5}_0} = 0$, which reside in the representations $\mathbf{10} \otimes \mathbf{5} = \mathbf{35} \oplus \mathbf{10} \oplus \mathbf{5}$. However, the symmetric product of three moment map operators $\mu_{\mathfrak{c}_2}$ does not contain the representation $\mathbf{5}$, hence it needs to be added back in the plethystic logarithm. Additionally, the relations at $R = 2$ are a consequence of the saturation of the unitarity bounds of [33, 47], as was also observed in [19].

### 3.2.2 The $C_2 U_1$ vertex operator algebra

The strong generators of the VOA associated with the rank-one $C_2 U_1$ theory are in one-to-one correspondence with the Higgs branch chiral ring generators. Indeed, the singlet Higgs branch relation at $R = 2$ (see (3.31)) is tantamount to the statement that $c_{2d} = c_{\text{Sug,tot}}$ and so the conformal stress tensor is identified with the Sugawara stress tensor. This accounts for the $+qt$ contribution in the plethystic logarithm of the the Macdonald index. Table 6 lists the complete set of strong generators.

Table 6: Strong generators of the vertex operator algebra associated with the rank-one $C_2 U_1$ theory. The $\mathfrak{c}_2$ adjoint representation $\mathbf{10}$ is realised by a symmetrised pair of fundamental indices, while the $\mathbf{5}$ is realised as an $\Omega$-traceless, antisymmetrised pair of fundamental indices.

| $\mathcal{O}$ | $\mathbb{V}(\mathcal{O})$ | $h_{\mathbb{V}(\mathcal{O})}$ | $\mathfrak{c}_2 \times \mathfrak{u}(1)$ |
|---|---|---|---|
| $\mu_{\mathfrak{u}(1)}$ | $j$ | $1$ | $\mathbf{1}_0$ |
| $\mu_{\mathfrak{c}_2}$ | $\mathcal{J}_{IJ}$ | $1$ | $\mathbf{10}_0$ |
| $w$ | $W_{IJ}$ | $3/2$ | $\mathbf{5}_3$ |
| $\tilde{w}$ | $\widetilde{W}_{IJ}$ | $3/2$ | $\mathbf{5}_{-3}$ |

Bootstrapping the vertex operator algebra follows the same template as in the previous subsection. Denoting by $\Omega$ the symplectic form of $C_2$, we have the following canonical OPEs involving the affine currents,

$$j(z)\, j(0) \sim \frac{-9}{z^2} , \tag{3.39}$$

$$\mathcal{J}_{IJ}(z)\, \mathcal{J}_{KL}(0) \sim \frac{-2\,\Omega_{L(I}\Omega_{J)K}}{z^2} + \frac{2\,\Omega_{(I(K}\,\mathcal{J}_{J)L)}(0)}{z} , \tag{3.40}$$

$$j(z)\, W_{IJ}(0) \sim \frac{+3\, W_{IJ}(0)}{z}\,, \tag{3.41}$$

$$j(z)\, \widetilde{W}_{IJ}(0) \sim \frac{-3\, \widetilde{W}_{IJ}(0)}{z}\,, \tag{3.42}$$

$$\mathcal{J}_{IJ}(z)\, W_{KL}(0) \sim \frac{2\, \Omega_{(I[K}\, W_{J)L]}(0)}{z}\,, \tag{3.43}$$

$$\mathcal{J}_{IJ}(z)\, \widetilde{W}_{KL}(0) \sim \frac{2\, \Omega_{(I[K}\, \widetilde{W}_{J)L]}(0)}{z}\,, \tag{3.44}$$

and the additional non-zero OPE is $W \times \widetilde{W}$, which is given by

$$\begin{aligned}
W_{K_1 L_1}(z)\, \widetilde{W}_{K_2 L_2}(0) \sim\ & \frac{\Delta^{\mathbf{5,5}}_{K_1 L_1; K_2 L_2}}{z^3} + \frac{1}{z^2}\Big( -\frac{1}{3}\, \Delta^{\mathbf{5,5}}_{K_1 L_1; K_2 L_2}\, j + 2\, \Omega_{[K_1[K_2}\, \mathcal{J}_{L_1]L_2]} \Big) \\
& + \frac{1}{z}\Big( -\frac{1}{6}\, \Delta^{\mathbf{5,5}}_{K_1 L_1; K_2 L_2}\, \partial j + \Omega_{[K_1[K_2}\, \partial \mathcal{J}_{L_1]L_2]} + \frac{1}{18}\, \Delta^{\mathbf{5,5}}_{K_1 L_1; K_2 L_2}\, (jj) \\
& \quad - \frac{2}{3}\, \Omega_{[K_1[K_2}\, (j\mathcal{J}_{L_1]L_2]}) + \frac{1}{10}\, \Delta^{\mathbf{5,5}}_{K_1 L_1; K_2 L_2}\, \Omega^{PQ}\, \Omega^{RS}\, (\mathcal{J}_{PR}\mathcal{J}_{QS}) \\
& \quad + \frac{2}{3}\big( (\mathcal{J}_{[K_1[K_2}\mathcal{J}_{L_1]L_2]})|_{\Omega\text{-traceless}} \big) \Big)\,, \tag{3.45}
\end{aligned}$$

where

$$\Delta^{\mathbf{5,5}}_{K_1 L_1; K_2 L_2} := \Omega_{K_1 K_2}\Omega_{L_1 L_2} + \Omega_{K_1 L_2}\Omega_{K_2 L_1} - \frac{1}{2}\Omega_{K_1 L_1}\Omega_{K_2 L_2}\,, \tag{3.46}$$

and we have omitted the positional arguments for the operators on the right-hand side, which are all evaluated at the origin. The stress energy tensor is provided by the Sugawara construction for the full flavour symmetry, and is thus given by

$$T(z) = -\frac{1}{18}(jj)(z) - \frac{1}{2}\, \Omega^{PQ}\, \Omega^{RS}\, (\mathcal{J}_{PR}\mathcal{J}_{QS})(z)\,, \tag{3.47}$$

with the expected central charge $c_{2d} = -19$.

Null relations can be easily derived. At conformal weight two we find a relation in the representation $\mathbf{5}_0$,

$$0 = (\mathcal{J}_{L[K}\, \mathcal{J}_{M]N})\Omega^{LN} + \frac{1}{4}\Omega_{KM}\Omega^{PQ}(\mathcal{J}_{PL}\, \mathcal{J}_{QN})\Omega^{LN}\,. \tag{3.48}$$

At weight $h = \frac{5}{2}$, one computes

$$0 = -(\mathcal{J}_{K[I}\, W_{J]L})\Omega^{KL} + \frac{1}{4}\Omega_{IJ}\Omega^{MN}(\mathcal{J}_{MK}\, W_{LN})\Omega^{KL} + \frac{1}{6}(j\, W_{IJ}) + \frac{1}{2}\partial W_{IJ}\,, \tag{3.49}$$

$$0 = \big( (\mathcal{J}_{IK}\, W_{LJ}) + (\mathcal{J}_{JK}\, W_{LI}) \big)\Omega^{KL}\,, \tag{3.50}$$

transforming in $\mathbf{5}_{+3}$ and $\mathbf{10}_{+3}$, respectively. Their conjugates are of course also null. Next, at $h = 3$ one can verify the following null relations,

$$0 = (W_{IJ}W_{KL})\Omega^{IK}\Omega^{JL}\,, \tag{3.51}$$

$$\begin{aligned}
0 =\ & (W_{KL}\widetilde{W}_{MN})\Omega^{KM}\Omega^{LN} + \frac{5}{81}(jjj) - \frac{5}{9}(\partial jj) + \frac{1}{3}(\mathcal{J}_{KL}\, \mathcal{J}_{MN}\, j)\Omega^{KM}\Omega^{LN} \\
& - (\mathcal{J}_{KL}\, \partial \mathcal{J}_{MN})\Omega^{KM}\Omega^{LN} + \frac{5}{9}\partial^2 j\,, \tag{3.52}
\end{aligned}$$

$$\begin{aligned}
0 =\ & \frac{1}{18}(\mathcal{J}_{KL}\, jj) - (W_{M(K}\, W_{L)N})\Omega^{MN} + \frac{2}{9}(\mathcal{J}_{(KM}\, \mathcal{J}_{NO}\, \mathcal{J}_{PL)})\Omega^{MN}\Omega^{OP} \\
& + \frac{2}{9}(\mathcal{J}_{KL}\, \mathcal{J}_{MN}\, \mathcal{J}_{OP})\Omega^{MO}\Omega^{NP}\,, \tag{3.53}
\end{aligned}$$

$$0 = \frac{2}{3}(\mathcal{J}_{(KM}\,\mathcal{J}_{NO}\,\mathcal{J}_{PL)})\Omega^{MN}\Omega^{OP} + \frac{1}{6}(\mathcal{J}_{KL}\,\mathcal{J}_{MN}\,\mathcal{J}_{OP})\Omega^{MO}\Omega^{NP}\,, \tag{3.54}$$

$$0 = (W\,\widetilde{W})|_{\mathbf{14}_0} + \frac{2}{9}(j\,\mathcal{J}\,\mathcal{J})|_{\mathbf{14}_0} - \frac{2}{3}(\mathcal{J}\,\partial\,\mathcal{J})|_{\mathbf{14}_0}\,. \tag{3.55}$$

## 3.3 Theory 3: rank-two $\mathfrak{su}(3)$-instanton SCFT

So far, the identification of twisted $A_2$ theories has not been particularly surprising, with one entry previously studied and the second a new theory. This changes as we turn our attention to the next theory of the twisted $A_2$ family. To start with the conclusion, we find that

$$\text{rank-two } \mathfrak{su}(3)\text{-instanton SCFT} \quad \longleftrightarrow \quad \boxed{\phantom{xxx}}. \tag{3.56}$$

See, for example, [28] for a review of some properties of the rank-two $\mathfrak{su}(3)$-instanton SCFT, which we will also sometimes denote as $\mathcal{T}^{(2)}_{\mathfrak{su}(3)}$. The surprising feature of this identification was mentioned already in Section 1. It has often been asserted that only in the presence of irregular (wild) punctures can non-integer Coulomb branch dimensions occur, but the Coulomb branch spectrum of $\mathcal{T}^{(2)}_{\mathfrak{su}(3)}$ is $\Delta_1 = \frac{3}{2}, \Delta_2 = 3$. Comparing to the Coulomb branch spectrum of $\tilde{T}_3$, we see that apparently, changing a twisted trivial embedding into $\mathfrak{c}_1$ to a principal embedding replaces a dimension-three generator from the Coulomb branch spectrum with a dimension $\frac{3}{2}$ generator.

We now provide evidence for the identification (3.56). First, the Weyl anomaly coefficients, once again computed using (2.2), match those of the rank-two $\mathfrak{su}(3)$-instanton SCFT,

$$a = \frac{47}{24}\,, \qquad c = \frac{13}{6}\,. \tag{3.57}$$

Second, the Macdonald index

$$I_M(q,t;\mathbf{a},\mathbf{b}) = \sum_\lambda \frac{\text{PE}\big[\frac{t}{1-q}\chi^{\mathfrak{a}_2}_{\text{adj}}(\mathbf{a})\big]P^{\mathfrak{a}_2}_{(\lambda,\lambda)}(\mathbf{a})\,\text{PE}\big[\frac{t}{1-q}\chi^{\mathfrak{c}_1}_{\text{adj}}(\mathbf{b})\big]P^{\mathfrak{c}_1}_{(\lambda)}(\mathbf{b})\,\text{PE}\big[\frac{t^2}{1-q}\big]P^{\mathfrak{c}_1}_{(\lambda)}(t^{\frac{1}{2}},t^{-\frac{1}{2}})}{\text{PE}\big[\frac{t^2+t^3}{1-q}\big]P^{\mathfrak{a}_2}_{(\lambda,\lambda)}(t,1,t^{-1})}$$

$$= \text{PE}\bigg[\frac{1}{1-q}\Big\{\big(\chi^{\mathfrak{a}_1}_{\text{adj}}(\mathbf{b})+\chi^{\mathfrak{a}_2}_{\text{adj}}(\mathbf{a})\big)t + \chi^{\mathfrak{a}_1}_{\mathbf{2}}(\mathbf{b})\chi^{\mathfrak{a}_2}_{\text{adj}}(\mathbf{a})t^{\frac{3}{2}} - t^2 + qt - \chi^{\mathfrak{a}_1}_{\mathbf{2}}(\mathbf{b})\big(1+2\chi^{\mathfrak{a}_2}_{\text{adj}}(\mathbf{a})\big)t^{\frac{5}{2}}$$

$$+ \chi^{\mathfrak{a}_1}_{\mathbf{2}}(\mathbf{b})qt^{\frac{3}{2}} - \big(1+\chi^{\mathfrak{a}_1}_{\text{adj}}(\mathbf{b})+\chi^{\mathfrak{a}_1}_{\text{adj}}(\mathbf{b})\chi^{\mathfrak{a}_2}_{\text{adj}}(\mathbf{a})+\chi^{\mathfrak{a}_2}_{\text{adj}}(\mathbf{a})+\chi^{\mathfrak{a}_2}_{\mathbf{10}}(\mathbf{a})+\chi^{\mathfrak{a}_2}_{\overline{\mathbf{10}}}(\mathbf{a})\big)t^3+\dots\Big\}\bigg]\,, \tag{3.58}$$

shows that the trinion does not contain free hypermultiplets and that the flavour symmetry is

$$G_F = SU(2)_4 \times SU(3)_6\,, \tag{3.59}$$

as expected. The class $\mathcal{S}$ realisation immediately confirms the result of [32] that the $SU(2)_4$ factor carries a Witten anomaly. Third, its Hall-Littlewood index—the $q \to 0$ limit of (3.58)— agrees with the Hilbert series of $\mathcal{M}^{(2)}_{\mathfrak{su}(3)}$, the centred instanton moduli space of two $\mathfrak{su}(3)$ instantons, *i.e.*, the Higgs branch of $\mathcal{T}^{(2)}_{\mathfrak{su}(3)}$, as computed in [65,66] and whose generators and relations were decoded already in [28]. Fourth, its Schur limit ($t \to q$) agrees perfectly with the expression found in [28] (see equation (5.4) in that paper).[24] The vertex operator algebra

---

[24]See also (5.26) of [67], and (4.10) of [32], after identifying $\mathcal{T}_X$ of that reference with $\mathcal{T}^{(2)}_{\mathfrak{su}(3)}$ [28]. Note also that, taking a cue from [59] (where the TQFT-expression of the indices of (untwisted) trinion theories was reorganised in terms of characters of the critical-level current algebras residing at the punctures), reference [32] proposed an expansion of the $\mathcal{T}^{(2)}_{\mathfrak{su}(3)}$ Schur index in terms of critical characters, see their (7.1). This result can now be easily recognised and understood as simply being the TQFT-expression of the index of the twisted trinion theory (3.56).

$\mathcal{V}^{(2)}_{\mathfrak{su}(3)}$ of this theory was constructed in [28, 32]. It is strongly generated by the operators descending from the Higgs branch chiral ring generators—the critical affine $\mathfrak{a}_1 \times \mathfrak{a}_2$ currents and an $h = \frac{3}{2}$ generator transforming in the representation $(\mathbf{2}, \mathbf{3})$ of $\mathfrak{a}_1 \times \mathfrak{a}_2$—together with the Virasoro stress tensor.

## 3.4 Theory 4: $(A_1, D_4)$ Argyres-Douglas theory plus hypermultiplet

Our next entry will be the trinion . If we apply the tentative lesson learned below (3.56) about the change in the Coulomb branch spectrum that arises by making the replacement , but now starting with the rank-one $C_2 U_1$ theory, we expect this theory to be a rank-one theory whose unique Coulomb branch chiral ring generator has scaling dimension $\Delta = \frac{3}{2}$. According to the classification of rank-one theories of [64], there exists just one theory that fits the bill: the $(A_1, D_4)$ Argyres-Douglas theory. However, the trinion theory may contain additional hypermultiplets that, of course, do not contribute to the Coulomb branch spectrum. Indeed, we will verify momentarily that the trinion carries an additional single hypermultiplet, and thus we find

$$(A_1, D_4) \text{ Argyres-Douglas theory} \otimes \text{free hypermultiplet} \quad \longleftrightarrow \quad . \tag{3.60}$$

Once again we encounter a theory with fractional scaling dimensions for Coulomb branch chiral ring operators. As always, our first checks are the $a$ and $c$ central charges. We can easily compute them to be

$$a = \frac{5}{8} = \frac{7}{12} + \frac{1}{24}, \qquad c = \frac{3}{4} = \frac{2}{3} + \frac{1}{12}. \tag{3.61}$$

In the second equality, we have split off the contribution from a single free hypermultiplet $(a, c)_{\text{HM}} = (\frac{1}{24}, \frac{1}{12})$, and indeed recognise the central charges of the $(A_1, D_4)$ Argyres-Douglas theory.

The computation of the Macdonald index is by now familiar. Its result for the trinion of interest in this subsection is

$$I_M(q, t; a, \mathbf{b}) = \text{PE}\left[\frac{1}{1-q}\left\{\chi_2^{\mathfrak{a}_1}(\mathbf{b}) t^{\frac{1}{2}} + \chi_{\text{adj}}^{\mathfrak{a}_2}(a, \mathbf{b}) t - (1 + \chi_{\text{adj}}^{\mathfrak{a}_2}(a, \mathbf{b})) t^2 + qt + \dots \right\}\right]. \tag{3.62}$$

We spot a free hypermultiplet transforming as a doublet of $SU(2)_{\text{HM}}$ in the $t^{\frac{1}{2}}$ term, while the interacting part of the theory has flavour symmetry $SU(3)$, of which the punctures of the trinion only make manifest an $SU(2)_{\text{int}} \times U(1)$ subgroup, under which the fundamental representation decomposes as $\mathbf{3} \to \mathbf{2}_{-1} + \mathbf{1}_2$. More precisely, the trinion only makes manifest the diagonal symmetry group $SU(2)_{\text{diag}} = \text{diag}(SU(2)_{\text{HM}}, SU(2)_{\text{int}})$. The flavour central charge of $SU(2)_{\text{int}}$ can be deduced by observing that $SU(2)_{\text{diag}}$ has flavour central charge equal to four, while $SU(2)_{\text{HM}}$ has $k_{\text{HM}} = 1$. The flavour central charge of $SU(2)_{\text{int}}$ is thus three, which implies that the flavour central charge of $SU(3)$ is also three, as $SU(2)_{\text{int}}$ is embedded with embedding index one. Hence, for the flavour symmetry of the interacting part, we find $G_F = SU(3)_3$ as expected. The $SU(2)_{\text{HM}}$ factor (and thus also the $SU(2)_{\text{diag}}$ factor) carries a Witten anomaly. From the $q \to 0$ limit of (3.62) we find that the Higgs branch chiral ring of the interacting part of the theory is generated entirely by the moment map operators of this flavour symmetry. They satisfy the quadratic relations corresponding to the Joseph ideal, *i.e.*, $\mu^2\big|_1 = \mu^2\big|_{\text{adj}} = 0$. This is as it should be, since the Higgs branch of the $(A_1, D_4)$ Argyres-Douglas theory is the minimal nilpotent orbit of $SU(3)$, also known as the one-instanton moduli space of $SU(3)$

instantons. In fact, the values of the $c$ central charge and flavour central charge guaranteed these relations [33].

The vertex operator algebra associated with the $(A_1, D_4)$ Argyres-Douglas theory is simply the affine current VOA $\widehat{\mathfrak{su}(3)}_{-\frac{3}{2}}$ [33, 68, 69]. (The stress tensor is provided by the Sugawara construction as $c = c_{\text{Sug}}$). One can verify that the Schur index agrees to very high order in a series-expansion in $q$ with the vacuum character of this vertex operator algebra. It is useful for this comparison to recall that the latter quantity is given by [32, 70]

$$\chi_0(q; a, \mathbf{b}) = q^{\frac{1}{3}} \text{PE}\left[\frac{q}{1-q^2} \chi_{\text{adj}}^{\mathfrak{a}_2}(a, \mathbf{b})\right]. \tag{3.63}$$

The Casimir prefactor $q^{-c_{2d}/24}$ of the character is omitted in the Schur index, as the latter quantity is normalised to start with 1 in a series expansion.

## 3.5 Theory 5: two copies of $(A_1, D_4)$ Argyres-Douglas theory

The final member of the twisted $A_2$ family turns out to decompose into a tensor product theory,

$$\left((A_1, D_4) \text{ Argyres-Douglas theory}\right)^{\otimes 2} \quad \longleftrightarrow \quad \left( \vcenter{\hbox{⬚⬚  ▮⋮▮}} \right). \tag{3.64}$$

The usual checks can be performed. First, from our rule for understanding reduced twisted $A_2$ punctures we expect this trinion to describe a theory with two Coulomb branch chiral ring generators of equal scaling dimension $\Delta_1 = \Delta_2 = \frac{3}{2}$, which indeed is compatible with our identification above. Second, the conformal anomaly coefficients, computed using (2.2), agree with the proposal,

$$a = 2 \times \frac{7}{12}, \qquad c = 2 \times \frac{2}{3}. \tag{3.65}$$

Third, the Macdonald index can be massaged into the form

$$I_M(q, t; \mathbf{a}) = \text{PE}\left[\frac{1}{1-q}\left\{\chi_{\text{adj}}^{\mathfrak{a}_2}(\mathbf{a})t - (1 + \chi_{\text{adj}}^{\mathfrak{a}_2}(\mathbf{a}))t^2 + qt + \dots\right\}\right]^2, \tag{3.66}$$

which manifestly captures the double copy structure of the theory and shows that the global symmetry is enhanced to

$$G_F = SU(3)_3 \times SU(3)_3. \tag{3.67}$$

The punctures of the trinion only make manifest the diagonal subgroup of the full flavour symmetry. Each factor of the Macdonald index correctly matches that of the $(A_1, D_4)$ Argyres-Douglas theory. See the previous subsection for some more details. Finally, we identify the vertex operator algebra for this trinion theory as two commuting level $-3/2$ affine $\mathfrak{a}_2$ current algebra, and indeed, to very high order in $q$, the Schur limit of (3.66) matches the product of two vacuum characters presented in (3.63).

# 4 Interconnections between twisted $A_2$ trinions

So far, we have analysed each of the five members of the twisted $A_2$ family individually. However, the class $\mathcal{S}$ realisations of these theories reveal interesting interconnections between

them, which are established by performing partial Higgsings.[25] Indeed, partially closing a (full) puncture, *i.e.*, replacing a puncture specified by the trivial embedding with one labelled by some other embedding, can be implemented field-theoretically by giving a nilpotent vacuum expectation value to the moment map for the flavour symmetry carried by the full puncture and flowing to the IR [5,25–27]. In Table 1, all horizontally or vertically neighbouring theories are related to one another in this manner. In this section we explore these relations in some more detail. In doing so, we will recover instances of the more general framework developed in [28–30]. In particular, these references exploit and fully bring to bear the insights of [31] that these types of interrelation via Higgsing of a theory $\mathcal{T}_{\text{UV}}$ and a theory $\mathcal{T}_{\text{IR}}$ suggest a free-field realisation of the vertex algebra $\mathbb{V}(\mathcal{T}_{\text{UV}})$ in terms of the vertex algebra $\mathbb{V}(\mathcal{T}_{\text{IR}})$ together with additional free fields.

We start by considering the theory $\tilde{T}_3$ from Section 3.1. From the relations (3.5)–(3.12) we see that the Higgs branch of this theory, $\mathcal{M}_H^{\tilde{T}_3}$, contains the subvarieties

$$\mathcal{N}_{\mathfrak{su}(3)} \subset \mathcal{M}_H^{\tilde{T}_3} \ , \qquad \text{and} \qquad \mathcal{N}_{\mathfrak{su}(2)} \times \mathcal{N}_{\mathfrak{su}(2)} \subset \mathcal{M}_H^{\tilde{T}_3} \ , \tag{4.1}$$

where we set to zero $\{\omega, \mu_{\mathfrak{su}(2)}^{(1)}, \mu_{\mathfrak{su}(2)}^{(2)}\}$ and $\{\omega, \mu_{\mathfrak{su}(3)}\}$, respectively.[26] Here $\mathcal{N}_{\mathfrak{g}}$ is the nilpotent cone (of the complexification) of the Lie algebra $\mathfrak{g}$, *i.e.*, the subset of $\mathfrak{g}_{\mathbb{C}}$ containing all nilpotent elements.[27] Each nilpotent cone is associated to one of the full punctures. For example,

$$\mathcal{N}_{\mathfrak{su}(2)} = \left\{ \mu = \begin{pmatrix} p & e \\ f & -p \end{pmatrix} \, \middle| \, \text{tr}\,\mu^2 = 0 \right\} \cong \left\{ (p,e,f) \in \mathbb{C}^3 \, \middle| \, p^2 + ef = 0 \right\} \cong \mathbb{C}^2/\mathbb{Z}_2 \,. \tag{4.2}$$

The nilpotent cone $\mathcal{N}_{\mathfrak{g}}$ itself is stratified, with the strata being the nilpotent orbits $\mathcal{O}_e = G_{\mathbb{C}} \cdot e$ for $e$ a nilpotent element of $\mathfrak{g}_{\mathbb{C}}$. The Jacobson-Morozov theorem states that these are in one-to-one correspondence with conjugacy classes of embeddings $\Lambda : \mathfrak{su}(2) \to \mathfrak{g}$ via $e = \Lambda(\sigma^+)$. The reappearance of $\mathfrak{su}(2)$-embeddings is of course no coincidence. A partial Higgsing along $\mathcal{O}_{\Lambda(\sigma^+)} \subset \mathcal{N}_{\mathfrak{g}}$ is tantamount to partially closing a full puncture to one labelled by the embedding $\Lambda$ [5,25,26].

In the case at hand, we are interested in performing a partial Higgsing of $\tilde{T}_3$ along the nilpotent orbit $\mathcal{O}_{\Lambda_1(\sigma^+)} \subset \mathcal{N}_{\mathfrak{su}(3)}$, where $\Lambda_1 : \mathfrak{su}(2) \to \mathfrak{g}$ is the subregular embedding specified in Table 4a. This is the minimal nilpotent orbit. The class $\mathcal{S}$ realisation tells us that the theory flows in the infrared to the rank-one $C_2 U_1$ theory: $\left(\boxminus\,\square\right) \to \left(\boxdot\,\square\right)$. In particular, this implies that the vertex operator algebra we obtained in Section 3.2.2 is the subregular quantum Drinfel'd-Sokolov reduction of $\mathbb{V}(\tilde{T}_3)$ with respect to the $\widehat{\mathfrak{su}(3)}$ affine subalgebra [58]. The Higgs branch chiral ring of the rank-one $C_2 U_1$ theory can be similarly deduced from the one of $\tilde{T}_3$.

More intriguing are the partial Higgsing operations of $\tilde{T}_3$ along either one of the factors $\mathcal{N}_{\mathfrak{su}(2)} \times \mathcal{N}_{\mathfrak{su}(2)} \subset \mathcal{M}_H^{\tilde{T}_3}$. The nilpotent cone of $\mathfrak{su}(2)$ has just two strata: the origin and $\mathcal{N}_{\mathfrak{su}(2)} \setminus \{0\}$. The origin corresponds to performing no Higgsing at all. The nontrivial Higgsing of interest is thus triggered by a vacuum expectation value along the unique (nontrivial) nilpotent orbit of $\mathfrak{su}(2)$, *i.e.*, $\mathcal{N}_{\mathfrak{su}(2)} \setminus \{0\}$. For concreteness and without loss of generality, we choose an expectation value $\langle (\mu_{\mathfrak{su}(2)}^{(1)})_{++} \rangle \neq 0$. Our class $\mathcal{S}$ realisation indicates that the low-energy result of this Higgsing is the rank-two $\mathfrak{su}(3)$-instanton SCFT, $\left(\boxminus\,\square\right) \to \left(\boxminus\,\square\right)$. What's more, the type of Higgsing at work here has been analysed in detail in [28,29,31], and we can adapt

---

[25]It is tautological, but nevertheless important, to state that a partial Higgsing of the theory corresponds to giving a vacuum expectation to Higgs branch chiral ring operators corresponding to a point on a singular subvariety of the Higgs branch. This point of view was recently re-emphasised in [29,31,71]. The terminology is borrowed from the Lagrangian literature where partial Higgsings are characterised by the property that they do not fully Higgs the gauge group.

[26]The relations with representation-theoretic data in (3.10) are consistent with these statements.

[27]Nilpotent elements of $\mathfrak{g}_{\mathbb{C}}$ can be characterised by the requirement that all their Casimir invariants vanish.

the results obtained in those works to our current setting. In particular, we describe a dense, open set of the Higgs branch of $\tilde{T}_3$ as[28]

$$\mathcal{M}_H^{\tilde{T}_3} \supset \left(\widetilde{\mathcal{M}}_{\mathfrak{su}(3)}^{(2)} \times T^*(\mathbb{C}^*)\right)/\mathbb{Z}_2 \,. \tag{4.3}$$

Here $T^*(\mathbb{C}^*)$ has coordinates $(\mathrm{e}^{\frac{1}{2}}, \mathsf{h})$, while the coordinate ring of the centred two-instanton moduli space of $\mathfrak{su}(3)$ instantons $\widetilde{\mathcal{M}}_{\mathfrak{su}(3)}^{(2)}$ is generated by moment map operators $\tilde{\mu}_{\mathfrak{su}(2)}, \tilde{\mu}_{\mathfrak{su}(3)}$ of R-charge $R = 1$ and an additional generator $\tilde{\omega}$ of $R = 3$ that transforms as $(\mathbf{2}, \mathbf{8})$ under $\mathfrak{su}(2) \times \mathfrak{su}(3)$. The action of the $\mathbb{Z}_2$ quotient can be understood by noting that it should act as the centre of the $SU(2)$ flavour symmetry associated with the $\mathfrak{c}_1$-puncture that has been Higgsed. As explained in [72], this centre symmetry is identified amongst the different punctures, so in particular it is identified with that of the other $\mathfrak{c}_1$-puncture of $\tilde{T}_3$ which we can easily keep track of along the flow. The conclusion is that the $\mathbb{Z}_2$ acts by simultaneous negation of $\mathrm{e}^{\frac{1}{2}}$ and $\tilde{\omega}$. The generators of $\mathbb{C}[\mathcal{M}_H^{\tilde{T}_3}]$ can then be built from the ingredients present in the open set (4.3). For the moment map operators, we have

$$\left(\mu_{\mathfrak{su}(2)}^{(1)}\right)_{++} = \mathsf{e} \,, \qquad \left(\mu_{\mathfrak{su}(2)}^{(1)}\right)_{+-} = \frac{1}{2}\mathsf{h} \,, \qquad \left(\mu_{\mathfrak{su}(2)}^{(1)}\right)_{--} = \left(S^\natural - \frac{1}{4}\mathsf{h}^2\right)\mathsf{e}^{-1} \,, \tag{4.4}$$

where $S^\natural = -\frac{1}{2}(\tilde{\mu}_{\mathfrak{su}(2)})^2\big|_{\mathbf{1}}$, and, rather trivially,

$$\mu_{\mathfrak{su}(2)}^{(2)} = \tilde{\mu}_{\mathfrak{su}(2)} \,, \qquad \mu_{\mathfrak{su}(3)} = \tilde{\mu}_{\mathfrak{su}(3)} \,. \tag{4.5}$$

For the additional Higgs branch chiral ring generator $\omega$, we have

$$\omega_+ = \tilde{\omega}\, \mathrm{e}^{\frac{1}{2}} \,, \qquad \omega_- = \left(-\tilde{\mu}_{\mathfrak{su}(2)}\tilde{\omega}\big|_{(\mathbf{2}, \mathrm{adj})} - \frac{1}{2}\tilde{\omega}\mathsf{h}\right)\mathrm{e}^{-\frac{1}{2}} \,, \tag{4.6}$$

where the $\pm$ denotes the $\mathfrak{su}(2)^{(1)}$ index and other indices have been kept implicit. As in [28,31], this realisation of the Higgs branch chiral ring can be affinised with only moderate ingenuity. Doing so, one obtains a generalised free-field realisation of the vertex operator algebra $\mathbb{V}(\tilde{T}_3)$ in terms of two chiral bosons and the VOA $\mathcal{V}_{\mathfrak{su}(3)}^{(2)}$ associated with the two-instanton theory. This construction, together with its analogues for all two-instanton SCFTs of Lie algebras belonging to the Deligne series of exceptional Lie algebras, will be presented in [30]. Finally, the fact that Higgsing $\tilde{T}_3$ along the locus $\mathcal{N}_{\mathfrak{su}(2)} \times \mathcal{N}_{\mathfrak{su}(2)}$ does not decrease the dimensionality of the Coulomb branch indicates that $\tilde{T}_3$ in fact possesses a mixed branch over each point of its two-complex-dimensional Coulomb branch which intersects the Higgs branch precisely along the locus $\mathcal{N}_{\mathfrak{su}(2)} \times \mathcal{N}_{\mathfrak{su}(2)}$. This theory therefore has an *extended Coulomb branch* in the terminology of [64]. See [29,30] for more details.

Let us move on to analyse Higgsings of the rank-two $\mathfrak{su}(3)$-instanton SCFT. The class $\mathcal{S}$ picture indicates that its Higgs branch $\widetilde{\mathcal{M}}_{\mathfrak{su}(3)}^{(2)}$ contains the subvarieties

$$\mathcal{N}_{\mathfrak{su}(3)} \subset \widetilde{\mathcal{M}}_{\mathfrak{su}(3)}^{(2)} \,, \qquad \text{and} \qquad \mathcal{N}_{\mathfrak{su}(2)} \subset \widetilde{\mathcal{M}}_{\mathfrak{su}(3)}^{(2)} \,. \tag{4.7}$$

Interestingly, our class $\mathcal{S}$ realisation shows that Higgsing $\mathcal{T}_{\mathfrak{su}(3)}^{(2)}$ along the $\mathfrak{su}(3)$ nilpotent orbit inside $\mathcal{N}_{\mathfrak{su}(3)}$ labelled by the subregular embedding results in the $(A_1, D_4)$ Argyres-Douglas theory plus one free hypermultiplet. We expect similar Higgsings of two-instanton SCFTs to one-instanton SCFTs triggered by giving a vacuum expectation value to the moment map operator of the $\mathfrak{g}$-symmetry of the two-instanton SCFT to exist for all Lie algebras $\mathfrak{g}$ in the

---

[28]The cotangent bundle of $\mathbb{C}^*$ arises by solving the relation defining $\mathbb{C}^2/\mathbb{Z}_2$ for $f$ in the patch where $e \neq 0$. The coordinate $h$ then provides the cotangential direction.

Deligne series of exceptional Lie algebras. We leave a detailed study to future work. On the other hand, giving a Higgs branch vacuum expectation value along a nonzero point in $\mathcal{N}_{\mathfrak{su}(2)}$ in (4.7) has been studied in great detail in [28], and our class $\mathcal{S}$ pictures handily confirm the results obtained there. Indeed, we see that the Higgsed theory flows to two copies of the $(A_1, D_4)$ Argyres-Douglas theory: $\rightarrow$ . As a result, one can see that a dense open patch of $\widetilde{\mathcal{M}}^{(2)}_{\mathfrak{su}(3)}$ is given by

$$\widetilde{\mathcal{M}}^{(2)}_{\mathfrak{su}(3)} \supset \left( \widetilde{\mathcal{M}}^{(1)}_{\mathfrak{su}(3)} \times \widetilde{\mathcal{M}}^{(1)}_{\mathfrak{su}(3)} \times T^*(\mathbb{C}^*) \right)/\mathbb{Z}_2 \,. \tag{4.8}$$

The $\mathbb{Z}_2$ now acts by negating the $\mathbb{C}^*$ coordinate and exchanging the two copies of the one-instanton moduli space of $\mathfrak{su}(3)$ instantons, which we denoted by $\widetilde{\mathcal{M}}^{(1)}_{\mathfrak{su}(3)}$.[29] This realisation was similarly presented and successfully affinised in [28]. As above, this result indicates that the rank-two $\mathfrak{su}(3)$-instanton theory has an extended Coulomb branch that intersects the Higgs branch along $\mathbb{C}^2/\mathbb{Z}_2$.

Finally, we turn our attention to Higgsing the rank-one $C_2 U_1$ theory. In terms of the $\mathfrak{c}_1 \times \mathfrak{c}_1 \subset \mathfrak{c}_2$ that is manifest in the class $\mathcal{S}$ realisation, we would like to give one of the $\mathfrak{c}_1$ moment maps a nilpotent expectation value. The low-energy theory one obtains in this manner is predicted to be the $(A_1, D_4)$ Argyres-Douglas theory plus one hypermultiplet: $\rightarrow$ . By a parallel argument to that presented above, one can describe a dense open set of the Higgs branch of the rank-one $C_2 U_1$ theory according to

$$\mathcal{M}^{C_2 U_1}_H \supset \left( \widetilde{\mathcal{M}}^{(1)}_{\mathfrak{su}(3)} \times \mathbb{C}^2 \times T^*(\mathbb{C}^*) \right)/\mathbb{Z}_2 \,, \tag{4.9}$$

where the coordinate ring $\mathbb{C}[\mathcal{M}^{(1)}_{\mathfrak{su}(3)}]$ is generated by the $\mathfrak{su}(3)$ moment map operator subject to the Joseph relations. Like below (4.3), the $\mathbb{Z}_2$ acts according to the charge under the centre of the manifest $SU(2)$ symmetry of the infrared trinion. Upon decomposing the enhanced flavour symmetry $\mathfrak{su}(3)$ in terms of the flavour symmetry carried by the punctures as $\mathfrak{su}(3) \rightarrow \mathfrak{su}(2) \oplus \mathfrak{u}(1) \oplus \mathbf{2}_{+3} \oplus \mathbf{2}_{-3}$ (see below (3.62)), we see that the moment map operators associated with $\mathfrak{su}(2) \oplus \mathfrak{u}(1)$ are even, while those transforming in the representation $\mathbf{2}_{+3} \oplus \mathbf{2}_{-3}$ are odd under $\mathbb{Z}_2$. For the same reason, the $\mathbb{Z}_2$ also acts with a minus sign on the factor $\mathbb{C}^2$ describing the free hypermultiplet. Of course, it also negates the coordinate $e^{\frac{1}{2}}$ of $T^*(\mathbb{C}^*)$. All ingredients are then in place to construct $\mathbb{C}[\mathcal{M}^{C_2 U_1}_H]$ and to perform an affinisation of this realisation. This Higgsing is actually just one instance of a much more general analysis of interconnections between rank-one SCFTs studied in [29]. We refer the reader to that work for all details.

# 5 Dualities involving twisted $A_2$ theories

An elegant feature of theories of class $\mathcal{S}$ (briefly recounted in Section 2) is that their exactly marginal couplings are encoded as the complex structure moduli of their UV curve. Pair-of-pants decompositions of this Riemann surface correspond to weakly coupled gaugings of the trinion theories associated with the three-punctured spheres occurring in the decomposition,

---

[29]Note that the superconformal index can be enriched with a fugacity $s$ for the $\mathbb{Z}_2$ symmetry of our interest [72]. Its effect is most easily described in the TQFT expression for the index as a sum over representations $\lambda$ of $\mathfrak{c}_1$. It simply introduces a factor $s^{|\lambda|}$ in the summand, where $|\lambda|$ is the 2-ality of the representation, *i.e.*, the number of boxes in its Young diagram modulo two. Refining the index of "Theory 5" in this manner, one easily confirms the existence of an even and odd adjoint-valued operator with $R = 1$. The even operator is the diagonal combination of the currents; it arises from the $\lambda = 0$ term in the sum from the $K$-factor of the full puncture. The odd combination can then be identified as the difference of the two currents.

and so the distinct field theoretic descriptions of the different degeneration limits are all unified by a generalised $S$-duality. In this section we consider two paradigmatic examples of such generalised $S$-dualities involving the twisted $A_2$ theories presented earlier in the paper.

Let us start with the $\mathcal{N} = 2$ superconformal field theory described by an $SU(3)$ gauge theory with (generalised) matter given by three fundamental hypermultiplets and two copies of the $(A_1, D_4)$ Argyres-Douglas theory, which we have denoted earlier as $\mathcal{T}^{(1)}_{\mathfrak{su}(3)}$. This gauging is exactly marginal as the $\beta$-function vanishes, because the sum of the (four-dimensional) flavour central charges of the matter involved equals four time the dual Coxeter number of $SU(3)$: $k_{\text{tot}} = 6 + 3 + 3 = 12$. This theory was called $\mathcal{T}_{3,2,\frac{3}{2},\frac{3}{2}}$ in [73]. Recalling that the untwisted $A_2$ trinion with two full punctures and one minimal puncture encodes $3 \times 3$ free hypermultiplets and using the realisation of two copies of $\mathcal{T}^{(1)}_{\mathfrak{su}(3)}$ identified in (3.64), it is now obvious that this theory has a twisted $A_2$ class $\mathcal{S}$ realisation depicted below.

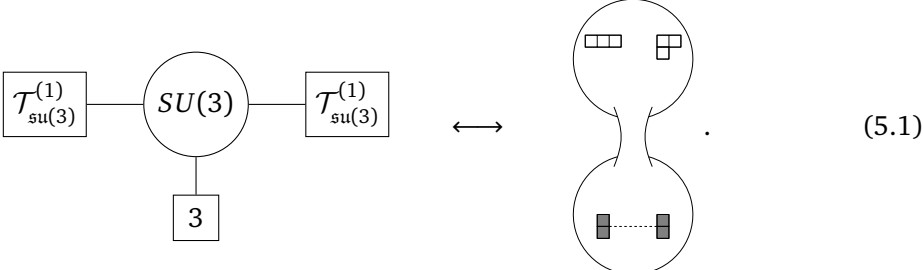

$$ \tag{5.1} $$

The gauging takes place along a full $A_2$ puncture. Upon driving the exactly marginal coupling constant to an strongly coupled cusp of the conformal manifold, a novel weakly-coupled description emerges. From the class $\mathcal{S}$ picture, we can identify the two alternative different degeneration limits of the surface, which are in fact equivalent. The result is shown in (5.2). The gauging takes place along a full twisted puncture, and using (3.60) and (3.56), we find an $S$-dual description in terms of both rank-one and rank-two $\mathfrak{su}(3)$-instanton theories.

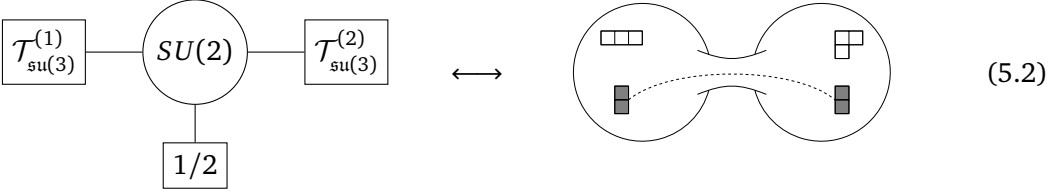

$$ \tag{5.2} $$

In words, the $S$-dual description of (5.1) is an $SU(2)$ gauge theory with one fundamental half-hypermultiplet, one copy of the $(A_1, D_4)$ Argyres-Douglas theory, of which an $SU(2)_3 \subset SU(3)_3$ flavour symmetry is gauged, and one copy of the rank-two $\mathfrak{su}(3)$-instanton theory, of which the $SU(2)_4$ flavour factor is gauged. It is easy to verify that this gauging is exactly marginal: $k_{\text{tot}} = 1 + 3 + 4 = 8 = 4 \times 2$. It is also clear that the Witten anomaly of the $SU(2)$ flavour symmetries of the half-hypermultiplet and the rank-two $\mathfrak{su}(3)$-instanton theory cancel. The study of precisely this $S$-duality relating (5.1) and (5.2) was the subject of the paper [32].[30] Our class $\mathcal{S}$ description confirms their findings and provide a particularly intuitive explanation of their result.

The second theory we consider is intricately self-dual. It is described by an $SU(2)$ gauge theory with one fundamental hypermultiplet and two copies of the $(A_1, D_4)$ Argyres-Douglas theory, of both of which an $SU(2)_3 \subset SU(3)_3$ subgroup is gauged. This gauging is exactly

---

[30]Their theory $\mathcal{T}_X$ was identified as the rank-two $\mathfrak{su}(3)$-instanton theory in [28], see also footnote 24. Note that in our class $\mathcal{S}$ realisation the extra hypermultiplet is incorporated into the trinion that also includes the $(A_1, D_4)$ Argyres-Douglas theory. In [32], the hypermultiplet was grouped together with the rank-two theory to give what the authors of [73] work dubbed the $\mathcal{T}_{3,\frac{3}{2}}$ theory.

marginal as $k_{\text{tot}} = 2 + 3 + 3 = 8 = 4 \times 2$. Before describing the class $\mathcal{S}$ realisation, let us recall from the canonical example of Argyres-Seiberg duality in class $\mathcal{S}$ that a degeneration limit of the Riemann surface of an $A_2$ theory that pinches off a pair of minimal punctures is described by one fundamental hypermultiplet coupled to an $SU(2)$ gauge theory that is also gauging an $SU(2) \subset SU(3)$ subgroup of the flavour symmetry associated with the full puncture along which the degeneration takes place [1].[31] With this insight, it is easy to see that we can give two different class $\mathcal{S}$ descriptions of the gauge theory of interest (in one of which the hypermultiplet is interpreted as two half-hypermultiplets), and that these descriptions are in fact $S$-dual to one another. The situation is illustrated in (5.3).

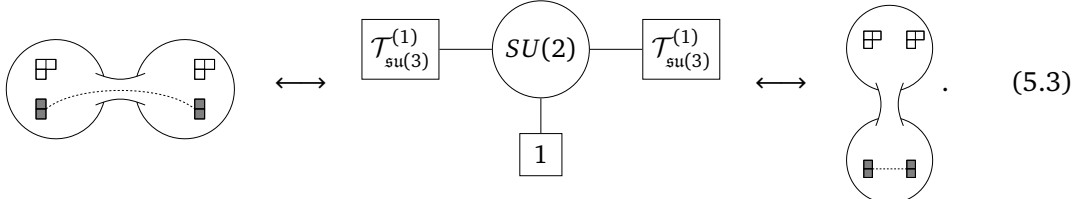

$$(5.3)$$

## 6 Future directions

We have seen that even the simplest family of twisted $A_{\text{even}}$ class $\mathcal{S}$ constructions contains a rich assortment of SCFTs, including (surprisingly) certain Argyres-Douglas type theories. This should be a good motivation for further study into this interesting corner of the class $\mathcal{S}$ biosphere. We conclude in this section with a couple of more specific comments about immediate extensions of this work.

### 6.1 Twisted $A_{2n}$ theories for $n > 1$

We see no obstruction in principle to directly pursuing an investigation of twisted $A_{2n}$ theories for $n > 1$ using the assorted diagnostics marshalled for the task in this paper. Though we leave such an examination for future work, to pique the reader's interest and to showcase the wealth of theories that remains to be uncovered amongst twisted $A_{2n}$ constructions,[32] we present here a realisation of the infinite series of theories dubbed $D_2[SU(2N + 1)]$, which were introduced in [74, 75] and of which the $(A_1, D_4)$ Argyres-Douglas theory is the $N = 1$ instance. The basic properties of the $D_2[SU(2N + 1)]$ theories are as follows:

- The $D_2[SU(2N + 1)]$ theory has rank $N$. Its Coulomb branch spectrum is $\Delta_i = \frac{2i+1}{2}$, $i = 1, \ldots, N$.

- The Weyl anomaly coefficients are given by $a = \frac{7}{24} N(1 + N)$ and $c = \frac{1}{3} N(1 + N)$.

- The flavour symmetry group is $SU(2N+1)$ with flavour central central charge $k = 2N+1$.

- The Schur limit of the superconformal index was conjectured in [70] to take the form

$$I_S(q; \mathbf{a}) = \text{PE}\left[\frac{q}{1 - q^2} \chi_{\text{adj}}^{\mathfrak{a}_{2N}}(\mathbf{a})\right]. \tag{6.1}$$

---

[31]In the parlance of [5–14] this involves an "irregular" puncture, though this should not be confused with our use of "irregular" elsewhere in this paper.

[32]In [19], a series of twisted $A_{2n}$ theories generalising the rank-one $C_2 U_1$ theory was analysed. These theories were dubbed $R_{2,2N}$ and arise by considering a trinion specified by two full twisted punctures and one untwisted puncture labelled by the subregular embedding (often called a minimal puncture).

- The associated vertex operator algebra is conjectured to be the affine current algebra [70]:

$$\mathbb{V}\big[D_2[SU(2N+1)]\big] = \widehat{\mathfrak{su}(2N+1)}_{-\frac{2N+1}{2}} \, . \tag{6.2}$$

Generalising the realisation (3.64) of the $(A_1, D_4)$ Argyres-Douglas theory plus one hypermultiplet, the above properties are reproduced (insofar as can be checked) by the identification

$$D_2[SU(2N+1)] \otimes (\text{free hypermultiplet})^{\otimes N} \quad \longleftrightarrow \quad \left(\begin{array}{c} [1^{2N}] \\ [N+1,N] \vdots \\ [2N] \end{array}\right), \tag{6.3}$$

where the untwisted $A_{2N}$ puncture is labelled by the embedding specified by the decomposition $\mathbf{2N+1} \rightarrow (\mathbf{N+1}) \oplus \mathbf{N}$ of the fundamental representation of $A_{2N}$ into $SU(2)$ representations, while one twisted puncture is labelled by the trivial embedding and the other one by the principal embedding. The Witten anomaly for the $USp(2N)$ symmetry at the full twisted puncture is carried by the hypermultiplets, while the full $SU(2N+1)$ flavour symmetry of the interacting part is an enhancement of the class $\mathcal{S}$ symmetries.

Furthering the analogy, two copies of the $D_2[SU(2N+1)]$ theory can be reproduced by the following twisted trinion,

$$(D_2[SU(2N+1)])^{\otimes 2} \quad \longleftrightarrow \quad \left(\begin{array}{c} [2N] \\ [1^{2N+1}] \vdots \\ [2N] \end{array}\right). \tag{6.4}$$

with only a diagonal subgroup of the two $SU(2N+1)$ symmetries made manifest. With these two trinions identifies, we can deduce a nice family of self-$S$-dualities involving the theories $D_2[SU(2N+1)]$ that generalises (5.3). The theory of interest is a $USp(2N)$ gauge theory with one fundamental hypermultiplet and two copies of the $D_2[SU(2N+1)]$ theory, of both of which a $USP(2N)_{2N+1} \subset SU(2N+1)_{2N+1}$ subgroup is gauged. Note that this gauging is exactly marginal as $k_{\text{tot}} = 2 + (2N+1) + (2N+1) = 4(N+1)$. In terms of the class $\mathcal{S}$ realisations given above, this gauge theory can be constructed as in the left-hand side of (6.5).

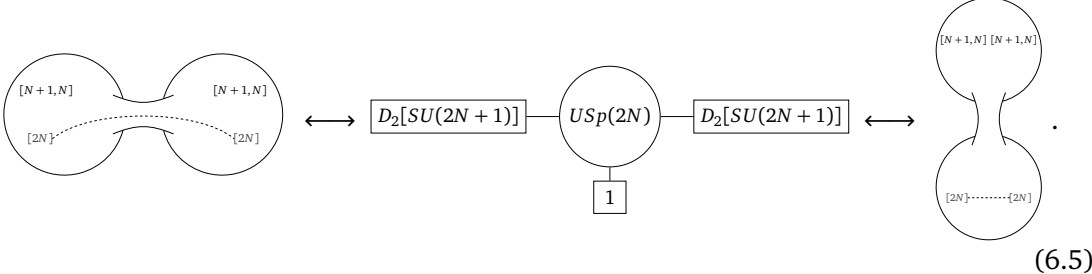

$$\tag{6.5}$$

The $S$-dual description is depicted on the right-hand side of (6.5), and involves colliding two $[N+1, N]$ punctures. In [6], such a degeneration of the UV curve was described explicitly for the cases $N = 1, 2$, and here we take the natural generalisation for granted. It amounts to a $USp(2N)$ gauge theory coupled to one fundamental hypermultiplet along with a $USp(2N) \subset SU(2N+1)$ symmetry associated with the full puncture along which the degeneration takes place. Thus the right-hand class $\mathcal{S}$ configuration again describes the same gauge theory.

Table 7: Proposal for pole structure and constraints for punctures arising in twisted $A_2$ theories. For the untwisted punctures this assignment is well-established. No constraints occur for untwisted $A$-type punctures. For the twisted punctures, this proposal correctly reproduces the scaling dimensions of all twisted $A_2$ theories, but at present lacks a first-principles derivation or independent verification.

| $\Lambda$ | $p_2(\Lambda)$ | $p_3(\Lambda)$ | constraints |
|---|---|---|---|
| ▭▭▭ | 1 | 2 | — |
| ⊞ | 1 | 1 | — |
| ▬ | 1 | $\frac{5}{2}$ | — |
| ▮ | 1 | $\frac{5}{2}$ | $a$-type of $d = 3$ |

## 6.2 Coulomb branch spectrum

In this work we have diligently avoided the question of how the Coulomb branch spectrum is encoded in twisted $A_{2n}$ theories. However, as we have emphasised above, the occurrence of half-integer Coulomb branch scaling dimensions is unexpected and deserving of further study. The conventional method to determine the spectrum, discussed and applied in great detail in [5–14] for all cases but twisted $A_{\mathrm{even}}$, involves two steps. We consider a (twisted) theory of class $\mathcal{S}$ of type j associated to a genus $g$ surface with punctures labelled by the embeddings $\Lambda_i$. Its Seiberg-Witten curve $\Sigma$ is the spectral curve of the associated Hitchin system. Hence, it can be written in terms of meromorphic $d_a$-differentials $\phi_{(d_a)}(z)$, where $d_a$, for $a = 1, \ldots, \mathrm{rank}\,j$ are the degrees of the Casimir invariants of j. The scaling dimension of $\phi_{(d_a)}$ is precisely $d_a$. The Hitchin field has prescribed singular boundary conditions at the location of each puncture. For regular (tame) punctures, the singularity is a well-understood simple pole, possibly with subleading fractional poles if the puncture is twisted. As a result, the differentials $\phi_{(d_a)}$ develop poles of order $p_{d_a}(\Lambda_i)$ at each of the punctures. The numbers $p_{d_a}(\Lambda)$ are called the pole structure of the puncture. Note that if the puncture is twisted these numbers may be fractional. The first step is then to compute the number of degrees of freedom in the meromorphic $d_a$-differential $\phi_{(d_a)}(z)$,

$$\sum_i p_{d_a}(\Lambda_i) + (g-1)(2d_a - 1). \tag{6.6}$$

This quantity gives a naive count of the dimension of the graded component of the Coulomb branch of degree $d_a$. Note that the scaling dimension $d_a$ is naturally integer. Apart from the global contribution $(g-1)(2d_a - 1)$, this expression suggests that each puncture adds $p_{d_a}(\Lambda)$ Coulomb branch operators of scaling dimension $d_a$; these can be identified with the coefficients of the poles of $\phi_{d_a}$ near the puncture. The reason the count is naive stems from the existence of constraints among the (leading) coefficients of the poles. The second step is to determine and take into account these constraints. Two types of constraints can occur. The so-called "$c$-type constraints" relate one coefficient to an expression in terms of other coefficients, effectively removing the operator corresponding to the original coefficient. It does not introduce new parameters. An "$a$-type constraint", on the other hand, states that a coefficient of scaling dimension $d$ can be expressed in terms of the square of a new coefficient of degree $d/2$. These $a$-type constraints thus provide a mechanism for the theory to acquire Coulomb branch chiral ring generators of scaling dimensions other than the degrees of the j invariants. In all cases in the literature, the initial scaling dimension $d$ is an even number.

A natural candidate for a mechanism to account for the half-integer scaling dimensions would thus be the existence of $a$-type constraints for coefficients of odd scaling dimension. While we leave an in depth exploration of this possibility for future work, we content ourselves here with pointing out that the pole structures and constraints proposed in Table 7 do indeed reproduce the scaling dimensions as reported in Table 1.

## Acknowledgements

The authors would like to thank Jacques Distler, Simone Giacomelli, Carlo Meneghelli, and Sujay Nair for helpful conversations and useful suggestions. This work was partially supported by grant #494786 from the Simons Foundation.

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
