# Peer review of "Argyres-Douglas Theories in Class S Without Irregularity"

_SciPost Physics, doi:SciPost Phys. 12, 172 (2022)_

## Round 1 · Referee Report · Anonymous (Referee 1) · 2022-2-26

Strengths
- The article contains new, interesting, and useful results.
- The results are stated clearly and explicitly, making the article useful as a reference.
- The material is well-presented.
Report
This article examined a number of theories of class $\mathcal{S}$ associated with a sphere with regular punctures. The authors demonstrated that certain superconformal field theories (SCFTs) of the Argyres-Douglas type can be engineered in this way. Moreover a new rank-two SCFT, dubbed the $\tilde{T}_3$ theory, was proposed by the authors of this article. Later it has been found that this theory can in fact be realized as a theory on two D3 branes probing an exceptional $E_6$ seven-brane singularity in the presence of an $\mathcal{S}$-fold without flux. For the theories explored in this article, their vertex operator algebras have been studied in-depth and the Schur indices have been computed. Since this article contains interesting and useful results, it deserves publication in the SciPost.

---

## Round 1 · Referee Report · Anonymous (Referee 2) · 2022-4-24

Report
The interpretation of the twisted $A_{2n} $ theories of class-S has proven somewhat mysterious. If one applies the standard formulae for the scaling dimensions of the Coulomb branch generator (along with the rest of the standard formulae that the authors claim are still correct), one rapidly runs into contradictions. The resolution, according to the present authors is that the $U(1)$ r-charges of the Coulomb branch generators are modified from the "naive" ones (where the r-charge of a meromorphic $k$-differential on C is $k$). Instead, the r-charges can be fractional, as in Argyres-Douglas theories.
In the usual formulation of Argyres-Douglas theories (on the sphere, where the Higgs field has an irregular singularity at a point), the modification is well-understood: the r-charge of the infrared SCFT is a linear combination of the "usual" one (which comes from the partial topological twist) and the U(1) isometry of the twice-punctured sphere. Here, the origin of the modification is more mysterious. Indeed, not just the r-charges, but even the number of Coulomb branch generators differs from the "naive" answer.
Clearly, while many mysteries remain, this paper represents a big step forward and should be published.
In the usual formulation of Argyres-Douglas theories (on the sphere, where the Higgs field has an irregular singularity at a point), the modification is well-understood: the r-charge of the infrared SCFT is a linear combination of the "usual" one (which comes from the partial topological twist) and the U(1) isometry of the twice-punctured sphere. Here, the origin of the modification is more mysterious. Indeed, not just the r-charges, but even the number of Coulomb branch generators differs from the "naive" answer.
Clearly, while many mysteries remain, this paper represents a big step forward and should be published.

---

## Editorial Decision

published